# UniVis: A Universal Framework for Computer Vision Tasks

## Abstract

We propose `UniVis`, a universal learning framework to tam a wide range of computer vision tasks, including visual understanding (e.g., semantic segmentation), low-level image processing (e.g., denoising), and conditional image generation (e.g., edge-to-image synthesis). Built on a large-scale pre-trained text-to-image diffusion model, `UniVis` unifies various vision tasks through a general framework using instruction tuning, where its unifying ability comes from the generative and reasoning power of the pre-trained model. Specifically, `UniVis` defines a general image completion task wherein the input consists of a pair of input-output images corresponding to the target task and a query image, and the aim is to generate the "missing" data paired to the query. The paired images play the role of image instruction defining the task, e.g., semantic segmentation is represented by an RGB image and its segmentation mask. Our rationale is that each computer vision task can be characterized by its unique input-output pair, which informs our `UniVis` model about the expected output for the given query. Furthermore, a task-level or instance-level prompt can be optionally added to provide text instruction. By unifying various visual tasks, `UniVis` has the advantage of minimizing the inductive bias inherent in designing models for individual tasks, and it also suggests that the understanding of different visual tasks can be achieved through a shared generative model. In experiments, `UniVis` showcases impressive performance on a bunch of standard computer vision benchmarks including ten tasks in total. The source code will be made publicly available.

## 1 Introduction

The natural language processing (NLP) community has witnessed a great success of large language models (LLMs) (Devlin et al., 2019; Radford et al., 2019; Raffel et al., 2020; Brown et al., 2020; Chowdhery et al., 2022) in recent years. A compelling advancement is that LLMs can serve as a generalist to handle a wide range of downstream tasks with a single general framework (Brown et al., 2020). This can be attributed to 1) emerging abilities brought by large-scale training (Wei et al., 2022a), 2) a unified task formulation (e.g., a variety of NLP tasks can be consistently framed as text completion (Brown et al., 2020)), and 3) in-context learning techniques that can help the model readily adapt to downstream tasks (Brown et al., 2020; Liu et al., 2021; Min et al., 2022; Rubin et al., 2021; Wei et al., 2021; Alayrac et al., 2022).

In the computer vision (CV) community, a unified framework for different tasks is also a long-standing aspiration. This is appealing because it side-steps task-specific designs, therefore minimizing the inductive bias inherent in devising models for individual tasks. However, the progress of such unification in CV lags behind NLP. There are three main Challenges. **C1**: Vision tasks encompass highly heterogeneous signals (e.g., RGB images, segmentation maps, and keypoints), impeding the unification of expert models for different tasks. **C2**: LLMs that undergo simple pre-training (e.g., masked language modeling and next-word prediction) exhibit superior linguistic understanding due to the semantic-dense nature of language created by humans. In contrast, most vision backbones trained via contrastive learning (Chen et al., 2020; He et al., 2020; Grill et al., 2020), masked image modeling (He et al., 2022; Xie et al., 2022), or generative modeling (Van Den Oord et al., 2017; Karras et al., 2019; Ho et al., 2020) still fall short of tackling various tasks within a unified model **C3**: It is convenient to incorporate in-context learning for NLP tasks (e.g., simply prepending a question-answer text for the mathematical reasoning task (Wei et al., 2022b)). It is, however, non-

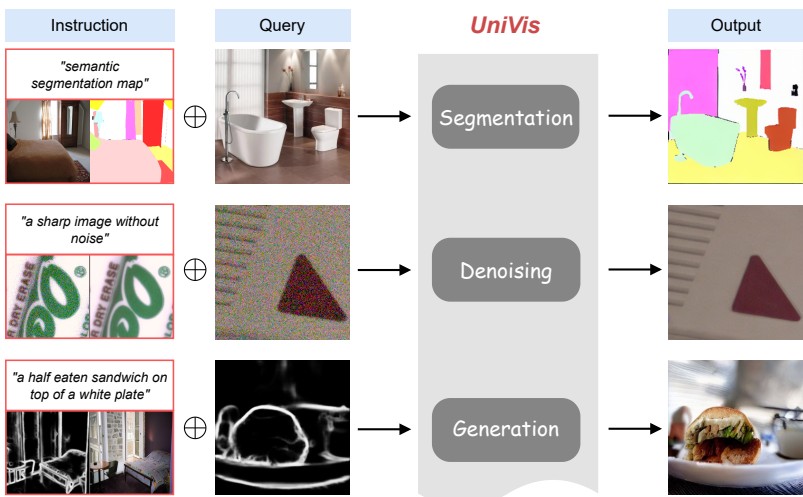

Figure 1: Illustration of the unifying capability of the proposed `UniVis`. `UniVis` can cope with different kinds of computer vision tasks within a universal framework, with each task instructed by an input-output image pair (and optionally a text prompt).

trivial to apply similar ideas directly in vision tasks. Therefore, how to cope with different vision tasks within a unified learning framework is still an open problem.

In this paper, we tackle the problem from the perspective of generative modeling, and introduce a universal learning framework called `UniVis`. `UniVis` unifies the learning processes of various vision tasks, including visual understanding (e.g., semantic segmentation), low-level image processing (e.g., denoising), and conditional image generation (e.g., edge-to-image synthesis), and can yield a unified vision model when jointly trained on these tasks. Specifically, to solve **C1**, `UniVis` copes with the heterogeneity in visual signals by formulating the input and output of different vision tasks as RGB images and defining a general image completion framework. The input of `UniVis` is a spatial concatenation of a query image and an input-output pair from the target task. The goal of `UniVis` is to generate the "missing" data paired to the query image in that task. Anticipating `UniVis` to yield promising results for various vision tasks in RGB image format, we favor vision backbones trained with generative modeling over commonly adopted pre-trained models like ViT (Dosovitskiy et al., 2021)-based MAE (He et al., 2022) or VQGAN (Esser et al., 2021) model, owning to the established excellence of generative models in generating high-quality RGB images. Among available generative models, a text-to-image diffusion model, Stable Diffusion (SD) (Rombach et al., 2022), is one of the very few trained on web-scale data LAION-5B (Schuhmann et al., 2022), which could provide us with a robust prior that incorporates a rich understanding of both visual and linguistic signals. We thus use SD as the backbone and fix its encoder and decoder when plugged into `UniVis` (which solves **C2**).

`UniVis` empowers SD to handle different vision tasks by devising an instruction tuning method (to solve **C3**), inspired by in-context learning in NLP. To achieve this, we introduce two key designs. First, similar to demonstration examples given to prompt the NLP task, we establish an input-output pair as the image instruction to characterize each vision task. For instance, semantic segmentation is represented by an RGB image and its segmentation masks. This instruction informs the model about what task it should execute for the query image. Moreover, we can optionally assign a task-level or instance-level prompt to provide text instruction and enjoy more flexible control over the results in the conditional image generation regime. Second, strong reasoning power is thought to be the reason why some LLMs are able to perform in-context inference on novel tasks (Wei et al., 2022a). In `UniVis`, we introduce an image completion framework that leverages the full potential of SD's reasoning abilities (Li et al., 2023a; Krojer et al., 2023), manifesting them in the form of image generation. Specifically, given an example input-output pair alongside a query image, the SD is designed to generate the corresponding "missing" data for the query, aligning with the task exemplified by the given pair.

Built on the above designs, `UniVis` serves as a unified framework for visual task learning, as shown in Figure 1, including but not limited to visual understanding, low-level image processing,

and conditional image generation. In practice, it can be employed to produce three types of models, based on the number and categories of given tasks. 1) When training on joint data in different categories of tasks (such as image generation, denoising, and semantic segmentation), a compact model can be derived, and it inherently possesses the capability to generate outputs for every given task. 2) When data from visual tasks of a specific category are aggregated for training, a single-category task model can be derived. For instance, consolidating data from both mask-to-image and depth-to-image results in a multifunctional generative model. 3) When data is from an individual task, it produces a dedicated model for that task. *We highlight that while these three types of `UniVis` trained in different regimes are based on different training datasets and hence yield distinct model parameters, the training approach remains exactly the same, underscoring the "universal" essence of our `UniVis`.* For evaluation, we showcase extensive results of these three types of models by conducting experiments on ten vision tasks in total. Intriguingly, we find that `UniVis` exhibits impressive performance on various vision tasks (including prediction and generation which are typically studied exclusively). This implies a potential of generative modeling in CV to embark on a similar trajectory as in NLP.

Our contribution is thus three-fold: 1) a universal learning framework that can cope with a wide range of computer vision tasks; 2) a new instruction tuning method that can be applied to the SD model, allowing its pre-trained knowledge to be adaptable to different kinds of downstream vision tasks; and 3) extensive experiments on a total of ten vision tasks and for three types of model training, by which we hope to spur more interesting research on how to induce a profound understanding of vision tasks through a shared scheme of generative modeling.

## 2 RELATED WORKS

**Unified Vision Models.** Encouraged by the success of language generalist models (Brown et al., 2020; Chowdhery et al., 2022; Touvron et al., 2023), seeking a generalist model for unifying different computer vision tasks has attracted significant interest in recent years. Some attempts (Wang et al., 2022a; Chen et al., 2022a; Kolesnikov et al., 2022; Chen et al., 2022b; Lu et al., 2023a) map the input image into discrete representations and implement prompt learning in the discrete space to deal with different tasks. A representative work, Unified IO (Lu et al., 2023a), homogenizes various vision data modalities into a sequence of discrete vocabulary tokens and utilizes VQGAN (Esser et al., 2021) to support dense prediction tasks. However, the discretization process causes lossy data compression, which is suboptimal for vision tasks. Uni-Perceiver series (Zhu et al., 2022b;a; Li et al., 2023c) introduce a unified maximum likelihood estimation pipeline for different modalities but they have not been verified in image generation tasks.

Another track of studies (Bar et al., 2022; Wang et al., 2022b; 2023b; Geng et al., 2023) utilizes the image as a general interface to unify vision tasks. MAE-VQGAN (Bar et al., 2022) and Painter (Wang et al., 2022b) use a masked image modeling solution, where the input image and an example pair are stitched together and the model only needs to predict the masked region. However, they demonstrate their validity in only image prediction tasks and have not been verified in other computer vision tasks like image generation. Concurrent with our work, InstructDiffusion (Geng et al., 2023) proposes a generalist model by casting different vision tasks as text-guided image editing. Despite the competitive performance, InstructDiffusion heavily relies on delicate training data construction and it does not support some of the vision tasks that are almost impossible to instruct by using human language (e.g., depth estimation). Another closely related method, PromptDiffusion (Wang et al., 2023b), incorporates in-context learning into a pre-trained diffusion model, enabling the integration of various vision-language tasks. PromptDiffusion sums up the features of context and query to perform the in-context modeling. However, context and query are not spatially aligned. The operation of feature addition would bring interference to the output, which may lead to suboptimal performance. In contrast, the proposed `UniVis` defines an image completion pipeline, which integrates context and query in a more reasonable way—spatial-wise concatenation where alignment is no longer required between context and query.

**Diffusion Models.** Diffusion models (Sohl-Dickstein et al., 2015; Ho et al., 2020; Song et al., 2020a;b; Song & Ermon, 2020) have recently become the primary choices for generative modeling of data. Following the definition in Denoising Diffusion Probabilistic Models (DDPM) (Ho et al., 2020), a diffusion model consists of a forward diffusion process that gradually adds noise to data

and a reverse denoising process that reconstructs desired data from the noise. Recent methods based on diffusion models achieve state-of-the-art results in many vision tasks, including image/video generation (Dhariwal & Nichol, 2021; Luo et al., 2023), image editing (Meng et al., 2021; Brooks et al., 2023), low-level image processing (Saharia et al., 2022b; Wang et al., 2023a), etc. Notably, large-scale text-to-image diffusion models show compelling ability (Nichol et al., 2021; Saharia et al., 2022a; Rombach et al., 2022; Balaji et al., 2022) to generate high-fidelity visual content. These pre-trained models are broadly utilized in applications such as concept customization (Gal et al., 2022; Ruiz et al., 2023) and image composition (Lu et al., 2023b). Some recent works (Li et al., 2023a; Clark & Jaini, 2023; Zhao et al., 2023; Xu et al., 2023; Tian et al., 2023) further reveal the potential of applying pre-trained diffusion models to discriminative tasks, which encourages us to build a universal framework that unifies generative and discriminative model training.

**Instruction Tuning.** GPT-3 (Brown et al., 2020) has demonstrated the ability of LLMs to perform various NLP tasks via language instructions. After that, there have been efforts in exploring the ability of instruction tuning (Mishra et al., 2021; Wei et al., 2021; Sanh et al., 2021). By fine-tuning language models on a collection of datasets described via instructions (e.g., task prompt, demonstration examples, and constraints), the model's generalization on unseen tasks obtains significant improvement (Wang et al., 2022c; Ouyang et al., 2022; Chung et al., 2022; Wu et al., 2023). Instruction tuning has recently been introduced to vision-language tasks, as well (Alayrac et al., 2022; Liu et al., 2023; Gao et al., 2023; Li et al., 2023b). A representative work, Flamingo (Alayrac et al., 2022), bridges pre-trained vision and language models by fine-tuning them on text-image instruction-following data and showcases impressive few-shot results in a variety of tasks such as image captioning and visual question-answering. By comparison, `UniVis` exploits a new approach of instruction tuning, which is based on the use of an image-label pair as well as an optional text for both image- and text-level instructions.

## 3 PROPOSED APPROACH

We propose `UniVis`, a universal framework that can solve various computer vision tasks. The aim of `UniVis` is to learn a mapping function $f$ that resembles instruction-following inference as:

$$f(E_{in}, E_{out}, y, I_{query}) = I_{gt}, \tag{1}$$

where $(E_{in}, E_{out})$ represents an example pair that serves as the image instruction to characterize a vision task ($E_{in}$ is the input and $E_{out}$ is the expected output, if learning a conventional model for that task). Taking semantic segmentation as an example, $E_{in}$ and $E_{out}$ represent an RGB image and its corresponding segmentation map, respectively. $y$ is a textual input acting as the text instruction to prompt the task and/or provide instance-level information (which is optional in practice). $I_{query}$ is the query image, and $I_{gt}$ is the corresponding ground truth in the task defined by the instruction.

To learn this function, we first construct instruction-based data for training, where the aim is to unify the input-output data formats for different vision tasks (Section 3.1). Then, on the top of a large-scale pre-trained diffusion model SD (Rombach et al., 2022), we devise an instruction tuning framework on which each time we can train with a batch of instruction data from different vision tasks (Section 3.2).

### 3.1 DATA CONSTRUCTION

We divide vision tasks into three categories: visual understanding, low-level image processing, and conditional image generation. In the following, we will introduce the specific tasks we focus on and elaborate on how to construct the instruction-based data using their conventional datasets. The main idea for construction is transforming all data formats into RGB images, by which we can implement spatial-wise concatenation of any input, out, and query samples (i.e., stitching all the images together into a grid as illustrated in Figure 2).

**Visual Understanding.** We conduct experiments on three representative prediction tasks, including semantic segmentation, depth estimation, and keypoint detection. Semantic segmentation is a dense classification task wherein the output is per-pixel semantic labels. We follow Painter (Wang et al., 2022b) to transfer these outputs to RGB images by assigning a color to each pixel according to a predefined semantic-color codebook, ensuring that each semantic class has its unique RGB value.

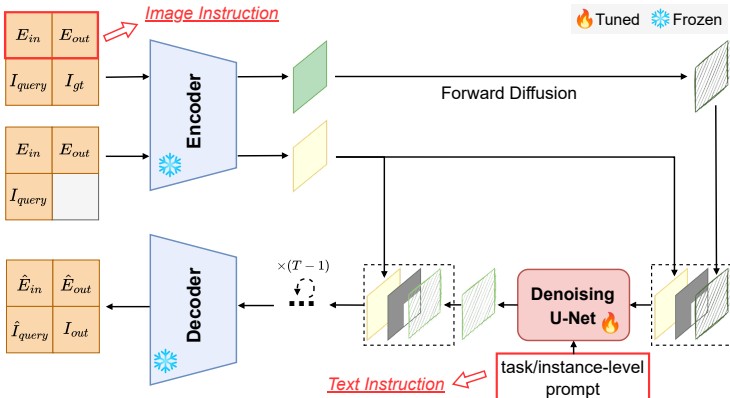

Figure 2: An overview of the proposed framework. It utilizes a pre-trained SD to perform image completion via instruction tuning. The ground truth is established as a grid image where each row is an input-output pair from the same task. The first row composed by $E_{in}$ and $E_{out}$ serves as the image instruction, and the model is trained to predict $I_{gt}$ paired to the query image $I_{query}$. At inference time, we crop out the lower right region of the infilled output as the final result $I_{out}$.

During inference, we obtain the predicted semantic class of each pixel by finding its "nearest" color in the codebook. Depth estimation is a dense regression task that needs to predict the depth value of each pixel of the input image. To convert the output (i.e., one-channel depth map) into an RGB image, we linearly scale the range of pixel values to $[0, 255]$ and replicate it three times to yield a three-channel result. Keypoint detection aims to locate key object components within the input image. We formulate the output as RGB images by drawing squares whose center is the location of keypoints and render each square with a unique color associated with the semantic class. [1]

To exploit the pre-trained knowledge of SD in understanding linguistic signals, we use textual prompts as text instructions. This prompt could be either *task-level* (e.g., "semantic segmentation map") or *instance-level* (e.g., "semantic segmentation map of a man sitting on a swing in a room"). In practice, the latter is obtained through an off-the-shelf image captioning tool (Li et al., 2022a).

**Low-level Image Processing.** We consider three typical tasks: image denoising, image deraining, and low-light image enhancement. The input and output of these tasks are RGB images, so we leave them unchanged to construct image instructions. Similar to visual understanding tasks, we devise two kinds of text instructions: 1) *task-level* prompt (e.g., "a sharp image without noise"), and 2) *instance-level* prompt (e.g., "a sharp image of a bathroom with a toilet, sink, and bathtub").

**Conditional Image Generation.** This line of tasks requires generating realistic images from conditions with sparse semantics, greatly differing from visual understanding and low-level image processing tasks. We consider four popular generation tasks in this paper, including mask-to-image, depth-to-image, pose-to-image, and edge-to-image. Inputs from the first three tasks can be converted to RGB format in the same way as used in visual understanding tasks. For the edge-to-image task, we adopt the edge detection model provided by ControlNet (Zhang et al., 2023) to generate HED edge maps (Xie & Tu, 2015) as the input. The captions of output images (e.g., "a cat sleeping on top of a pair of shoes") are used as text instructions.

## 3.2  INSTRUCTION TUNING FRAMEWORK

We implement `UniVis` as an instruction tuning framework on top of SD. SD is a text-to-image model that incorporates a diffusion process in the latent space of a pre-trained autoencoder. Specifically, a denoising U-Net is trained to fit the distribution of latent codes, which models the reverse diffusion process. Taking the noisy latent and the time step as input, this U-Net is further conditioned on the textual embeddings extracted through a text encoder CLIP (Radford et al., 2021) via cross-attention to produce the output at the current time step. During inference, SD performs iterative reverse diffusion on a randomly sampled noise to generate an image that faithfully adheres

---

[1]Due to space limits, the hyperparameters for drawing squares, rendering, etc., are given in the Appendix.

Table 1: Comparison results on visual understanding. *: Specialized methods for each task. ‡: Officially trained Painter model using $32\times$ the computing power of `UniVis`. †: Retrained using official code under the same computing resources as `UniVis`. **Bold**: Best. Underline: Second best. We ignore specialized models when ranking best and second best and this applies to all tables. The results of `UniVis` are reported as the average scores and standard deviations across three trials.

| Method | Segmentation | Depth estimation | | |
|---|---|---|---|---|
| | mIoU↑ | RMSE↓ | REL↓ | $\delta_1$↑ |
| OneFormer* (Jain et al., 2023) | 58.8 | - | - | - |
| Mask2Former* (Cheng et al., 2022) | 57.7 | - | - | - |
| ZoeDepth* (Bhat et al., 2023) | - | 0.270 | 0.075 | 0.955 |
| BinsFormer* (Li et al., 2022b) | - | 0.330 | 0.094 | 0.925 |
| Painter‡ (Wang et al., 2022b) | 49.9 | 0.288 | 0.080 | 0.950 |
| Painter† (Wang et al., 2022b) | 32.2 | **0.316** | **0.087** | **0.935** |
| PromptDiffusion (Wang et al., 2023b) | 18.2 | 0.746 | 0.171 | 0.799 |
| `UniVis-st` | **33.4** $\pm$ 0.4 | 0.420 $\pm$ 0.005 | 0.135 $\pm$ 0.004 | 0.857 $\pm$ 0.006 |

to the input text. To fulfill Eq. 1 when dealing with various tasks, we build an image completion pipeline and fine-tune the pre-trained SD using our prepared instruction-based data.

As shown in Figure 2, the image instruction (an example pair from a vision task, $E_{in}$ and $E_{out}$) is concatenated with another pair from the same task ($I_{query}$ and $I_{gt}$) to compose a grid image as the actual ground truth. During training, the input to the denoising U-Net comprises 3 components: 1) the noisy latent embedding of ground truth, 2) the latent embedding of a masked image $m$ similar to the ground truth but with $I_{gt}$ masked out, and 3) the binary mask $b$ indicating the masked region. The latter two serve as conditions to provide the model with context around the masked region and the location of the specific area to be infilled. Text instruction is sent to the text encoder and the extracted textual embeddings are injected into the denoising U-Net. With these instructions, the model is tuned to perform image completion, i.e., to generate the masked region. The training objective is the standard denoising loss of diffusion modeling:

$$\mathcal{L}(\theta) = \mathbb{E}_{z,m,b,y,\epsilon\sim\mathcal{N}(0,1),t}\left[\|\epsilon - \epsilon_\theta(z_t, t, \mathcal{E}(m), b, c_\phi(y))\|_2^2\right], \quad (2)$$

where $z$ is the latent code extracted from the ground truth, $y$ is the input text, $\epsilon$ is a noise term, $t$ is the time step, $\epsilon_\theta$ is the denoising U-Net, $z_t$ is the noisy version of $z$ at time $t$, $\mathcal{E}$ is the VAE encoder, and $c_\phi$ is the text encoder. We fine-tune the denoising U-Net while keeping the text encoder and the autoencoder of SD frozen.

Note that prior inpainting-based unified models (Bar et al., 2022; Wang et al., 2022b) apply masking to a portion of image patches. However, we argue that such patch-level inpainting that resembles word completion training schemes in NLP is not adequate for a holistic and profound understanding of vision tasks due to the fact that the correlation between pixels is much stronger than that between words (e.g., this redundancy presented in images makes the model readily inpaint a patch with neighboring patches). To mitigate this, we mask the *whole* desired output image and force the model to predict it during training. We will show later that this new strategy fosters a better connection between visual features and semantics. This finding is in line with that witnessed in MAE (He et al., 2022) where masking a very high portion of random patches facilitates more meaningful representation learning. It also implies an inherent difference between our generative modeling and the masked image modeling used by previous methods (Bar et al., 2022; Wang et al., 2022b).

## 4 EXPERIMENTS

**Datasets.** We conduct experiments on six datasets for ten vision tasks, including ADE20K (Zhou et al., 2017), NYUv2 (Silberman et al., 2012), COCO (Lin et al., 2014), Merged 5 datasets (Zamir et al., 2021), SIDD (Abdelhamed et al., 2018), and LoL (Wei et al., 2018). We adopt the same training/testing split as Wang et al. (2022b). Please refer to Table 6 for a detailed dataset configuration.

**Methods.** We evaluate `UniVis` with its two direct competitors, Painter (Wang et al., 2022b) and PromptDiffusion (Wang et al., 2023b), both designed to handle multiple tasks using a unified frame-

Table 2: Comparison results on low-level image processing. *: Specialized methods for each task. ‡: Officially trained Painter model using $32\times$ the computing power of UniVis. †: Retrained using official code under the same computing resources as UniVis. ⊎: Following InstructDiffusion (Geng et al., 2023), it directly reconstructs the ground truth via the autoencoder of pre-trained SD, and the corresponding results indicate the upper bound of UniVis. **Bold**: Best. Underline: Second best.

| Method | Deraining | | | Denoising | | | Enhancement | | |
|---|---|---|---|---|---|---|---|---|---|
| | PSNR↑ | SSIM↑ | LPIPS↓ | PSNR↑ | SSIM↑ | LPIPS↓ | PSNR↑ | SSIM↑ | LPIPS↓ |
| Restormer* (Zamir et al., 2022a) | 33.96 | 0.935 | 0.074 | 40.02 | 0.960 | 0.198 | - | - | - |
| MIRNet-v2* (Zamir et al., 2022b) | - | - | - | 39.84 | 0.959 | 0.203 | 24.74 | 0.851 | 0.116 |
| Painter‡ (Wang et al., 2022b) | 29.42 | 0.867 | 0.164 | 38.58 | 0.954 | 0.220 | 22.34 | 0.806 | 0.205 |
| Painter† (Wang et al., 2022b) | **25.84** | **0.840** | **0.191** | 32.84 | **0.933** | 0.224 | 20.18 | **0.733** | 0.354 |
| PromptDiffusion (Wang et al., 2023b) | 21.29 | 0.568 | 0.364 | 32.33 | 0.870 | 0.120 | 20.00 | 0.640 | 0.326 |
| UniVis-st | 22.62 | 0.598 | 0.302 | 34.55 | 0.907 | 0.095 | **20.63** | 0.681 | **0.256** |
| UniVis-sc | 22.64 | 0.599 | 0.301 | **34.80** | 0.910 | **0.092** | 19.91 | 0.665 | 0.286 |
| UniVis⊎ | 24.53 | 0.650 | 0.249 | 36.56 | 0.934 | 0.054 | 25.20 | 0.729 | 0.218 |

Table 3: Comparison results on conditional image generation. *: Specialized methods for each task. †: Trained using official code under the same computing resources as UniVis. Note that there is no officially trained Painter model for generation. **Bold**: Best. Underline: Second best.

| Method | Mask-to-image | Depth-to-image | Pose-to-image | Edge-to-image |
|---|---|---|---|---|
| | FID↓ | FID↓ | FID↓ | FID↓ |
| ControlNet* (Zhang et al., 2023) | 35.4 | 43.9 | 43.0 | 12.9 |
| Painter† (Wang et al., 2022b) | 75.7 | 89.3 | 200.1 | 233.1 |
| PromptDiffusion (Wang et al., 2023b) | 31.0 | 52.5 | 40.6 | 13.8 |
| UniVis-st | 29.9 ± 0.3 | **44.0 ± 0.7** | **34.7 ± 0.3** | 13.6 ± 0.2 |
| UniVis-sc | **27.8 ± 0.6** | 44.2 ± 0.8 | 34.3 ± 0.5 | **13.5 ± 0.4** |

work, as state-of-the-art methods. We also report the results of other competing methods, which are specially trained on single tasks and do not use a general framework, for reference purposes. Due to limited computing resources, we cannot jointly train UniVis on data from all tasks to achieve convergence in an affordable time. Therefore, we mainly report the results of single-task models (UniVis-st) that are separately trained for each task, and single-category models (UniVis-sc) that are jointly trained on data from multiple tasks of the same category. Nevertheless, we train a multi-category model (UniVis-mc) on data from three tasks belonging to distinct categories to demonstrate our UniVis's validity in tackling various tasks using a single set of model parameters.

**Implementation Details.** We utilize the same training settings of SD to optimize UniVis. We accumulate gradients every 16 batches with a batch size of 64. The learning rate is fixed to $6.4 \times 10^{-5}$. All training images are resized to $256 \times 256$ and we train UniVis on 4 RTX 3090 GPUs.

**Visual Understanding Results.** We assess the proposed UniVis on three visual understanding tasks described in Section 3.1. Standard metrics are adopted for evaluation: (1) mean Intersection-over-Union (mIoU) for semantic segmentation; (2) root mean squared error (RMSE), absolute relative error (REL), and the accuracy under the threshold ($\delta_1 < 1.25$) for depth estimation. Quantitative comparison results are presented in Table 1 and we make the following **O**bservations. **O1**: UniVis outperforms PromptDiffusion by a large margin, despite both adopting the pre-trained SD, albeit with significantly different frameworks. We attribute UniVis's superior performance to our more favorable image completion framework that integrates instructions spatially on the image-level rather than mixing them up on the feature-level. **O2**: The official results of Painter are better than ours, but Painter requires training on a considerable number of machines, i.e., around 128 RTX 3090 GPUs. Hence, we retrain Painter following its official code to compare with UniVis more fairly under the same compute. In this case, our UniVis are highly comparable with Painter. **O3**: UniVis as well as Painter and PromptDiffusion still fall behind most specialized methods, but the primary focus of this paper is to reveal the potential of generative modeling in building a universal solver for vision tasks, with achieving state-of-the-art performance being of our lower priority.

We also show some qualitative results of our method in Figure 3. UniVis succeeds in perceiving various scenes regarding semantics, depth, and salient object components, and subsequently

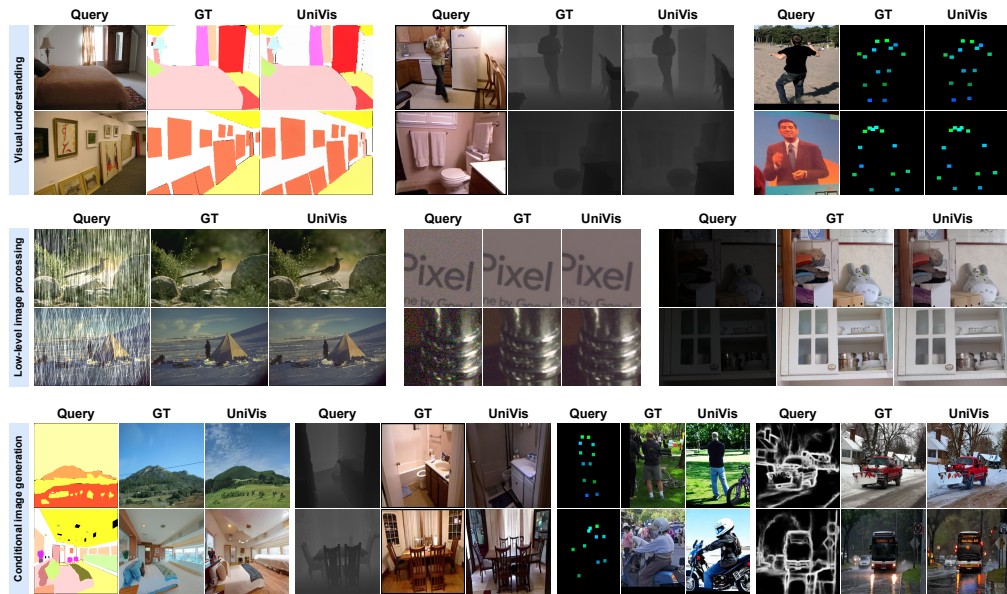

Figure 3: Visual results produced by our framework. Here we omit the instructions for simplicity. More visualizations are given in the Appendix due to space limits.

Table 4: Joint training results. We select three representative tasks from different categories.

| Method | Depth estimation | | | Denoising | | | Mask-to-image |
|---|---|---|---|---|---|---|---|
| | RMSE↓ | REL↓ | $\delta_1$↑ | PSNR↑ | SSIM↑ | LPIPS↓ | FID↓ |
| UniVis-st | 0.420 | 0.135 | 0.857 | 34.55 | 0.907 | 0.095 | 29.9 |
| UniVis-mc | 0.421 | 0.131 | 0.863 | 34.58 | 0.909 | 0.095 | 30.4 |

produces accurate predictions in RGB format. It is worth noting that UniVis performs well in keypoint detection (see Figure 3 and Figure 8 in the Appendix). Nevertheless, generating heatmaps to calculate metrics such as average precision (AP) is difficult to accomplish with the autoencoder of pre-trained SD as it introduces lossy compression. Limited by this, we do not report quantitative results. This issue can be alleviated by resorting to better pre-trained models in the future or employing an extra model to transfer the output to heatmaps as done in Geng et al. (2023).

**Low-level Image Processing Results.** We exploit the ability of UniVis to perform low-level image processing on three image restoration tasks. Standard metrics PSNR, SSIM, and LPIPS (Zhang et al., 2018) are used for evaluation. Table 2 presents the quantitative results of different methods. Similar to the observations in visual understanding, here UniVis attains competitive performance compared to Painter (retrained version) and surpasses PromptDiffusion in all metrics. In addition, there is an upper bound for UniVis because the autoencoder of pre-trained SD brings information loss (as pointed out in Geng et al. (2023)). We apply the autoencoder to reconstruct the ground truth and calculate the metrics as our upper bound. Visual results illustrated in Figure 3 also demonstrate the efficacy of UniVis in handling low-level image processing tasks.

**Conditional Image Generation Results.** We evaluate the conditional image generation performance of UniVis given various conditions, including segmentation mask, depth map, keypoint, and HED edge. The commonly used Fréchet Inception Distance (FID) (Heusel et al., 2017) is adopted to assess the realism of the generated images. The comparison results are reported in Table 3. **O**bservations are elaborated in the following. **O1**: The proposed UniVis achieves exceptional performance on all tasks and even surpasses the specialized method (ControlNet) on mask/depth/pose-to-image, indicating that UniVis fully unleashes the generative power of pre-trained SD. **O2**: Painter, which is built on top of pre-trained MAE, falls short of synthesizing realistic images from conditions with sparse semantics, resulting in poor FID values. Visual results of Painter shown in Figure 12, 13, 14, and 15 (Appendix) further verify this. **O3**: UniVis-sc attains a comparable performance to UniVis-st. This showcases the effectiveness of UniVis-sc in translating flexible control signals into high-fidelity images using a single model. As presented in

Table 5: Ablation study results for semantic segmentation on ADE20K.

| Method | Maskig strategy | | Type of text instruction | | |
|---|---|---|---|---|---|
| | region-wise | whole image | no prompt | task-level | instance-level |
| mIoU↑ | 17.4 | **33.4** | 31.0 | 31.1 | **33.4** |

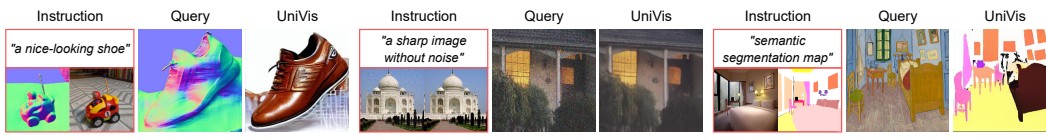

Figure 4: Example results from our `UniVis` in performing out-of-distribution inference.

Figure 3, given different conditions, `UniVis-sc` manages to recognize the underlying task and synthesize photorealistic images while spatially conforming to the control signals.

At last, we collect data from depth estimation, denoising, and mask-to-image to jointly train a multi-category model `UniVis-mc`. As shown in Table 4, `UniVis-mc` achieves a competitive performance very close to `UniVis-st`, confirming the proposed framework's ability to automatically identify the specific task through the given instructions and produce the corresponding desired output. It is encouraging to see the results of `UniVis-mc` trained for these tasks involving disparate visual signals and data domains, and we believe that unifying discrimination and generation will be made possible if the proposed `UniVis` can be trained with sufficient computational resources.

**Ablation study.** We perform ablations on two key ingredients of `UniVis`: the masking strategy and the design of text instruction. Instead of masking the whole image $I_{gt}$ during training, we randomly mask out a portion of $I_{gt}$ to train `UniVis` for the semantic segmentation task. As reported in Table 5, this region-wise masking results in a significant performance drop, highlighting the importance of our masking strategy in unleashing the unifying ability of pre-trained SD. We also study the effect of text instruction by training `UniVis` with three types of textual prompts, including no prompt (an empty string), task-level prompt, and instance-level prompt. We can find in Table 5 that instance-level prompt yields the best performance, which implies that detailed semantic information can facilitate the visual understanding ability of our model. Obtaining captions is convenient for visual understanding (using captioning tools) but manual construction is needed for other tasks. In practice, one needs to strike a balance between high-end performance and extra human efforts.

**Generalization capability.** We explore `UniVis`'s generalization capability by applying it to unseen tasks/data. As demonstrated in Figure 4, `UniVis` is capable of (1) generating realistic images from the normal map that is unseen during training, (2) denoising on in-the-wild images that have different data distribution compared to the training dataset, and (3) performing segmentation on images from new domains (e.g., Van Gogh's paintings in Figure 4(c)). These promising results indicate that `UniVis` learns the underlying "structure" of various visual signals and generalizes well to new scenarios by leveraging the pre-trained knowledge to conduct instruction-following inference.

## 5 CONCLUSION

In this paper, we explore the trajectory of LLMs to design a unified framework for computer vision tasks, identifying three essential components: 1) a general data interface for various tasks, 2) a powerful backbone with generative and reasoning ability, and 3) a visual instruction tuning method for efficient adaptation to various tasks. To this end, the proposed `UniVis` achieves the unification through a universal learning framework by 1) framing heterogeneous visual signals as RGB images, 2) leveraging the large-scale pre-trained Stable Diffusion (SD) as the backbone, and 3) introducing a new instruction tuning method based on image completion to adapt SD to different tasks. `UniVis`'s competitive performance across three categories of vision tasks verifies our design's potential of generative modeling in perceiving and processing visual signals in a general manner. Compared to the evaluated existing general frameworks, `UniVis` shows notable efficacy in handling at least one additional significant category of vision tasks, encouraging further research in this direction.

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

## A   DETAILED INFORMATION ON DATASETS

We adopt six widely employed datasets to evaluate the performance of different methods on ten computer vision tasks. A summary of dataset configuration is provided in Table 6. We give some further details in the following.

- ADE20K (Zhou et al., 2017) is a semantic segmentation dataset that has 20,210 images for training and 2,000 for validation, including 150 semantic classes. We conduct semantic segmentation and mask-to-image generation on this dataset.
- NYUv2 (Silberman et al., 2012) is a widely-used depth estimation dataset collected from indoor scenes. We adopt the dataset processed by Painter (Wang et al., 2022b), which contains 36,396 training images and 654 testing images. We conduct depth estimation and depth-to-image generation on NYUv2.

Table 6: Dataset configuration.

| Dataset | Task | Training images | Testing images |
|---|---|---|---|
| ADE20K (Zhou et al., 2017) | Segmentation | 20,210 | 2,000 |
| | Mask-to-image | 20,210 | 2,000 |
| NYUv2 (Silberman et al., 2012) | Depth estimation | 36,396 | 654 |
| | Depth-to-image | 36,396 | 654 |
| COCO (Lin et al., 2014) | Keypoint detection | 149,781 | 6,352 |
| | Pose-to-image | 149,781 | 6,352 |
| | Edge-to-image | 118,287 | 5,000 |
| Merged 5 datasets (Zamir et al., 2021) | Deraining | 13,712 | 4,300 |
| SIDD (Abdelhamed et al., 2018) | Denoising | 96,000 | 1,280 |
| LoL (Wei et al., 2018) | Enhancement | 485 | 15 |

- COCO (Lin et al., 2014) is a classical vision dataset which provides rich annotations of images such as segmentation masks, keypoints, and captions. We conduct keypoint detection and pose-to-image generation on COCO. Each human image is labeled with 17 keypoints. We also extract HED edge maps from images in COCO and perform edge-to-image generation task on those image-edge pairs.

- We conduct deraining, denoising, and low-light image enhancement on three benchmark datasets, namely Merged 5 datasets (Zamir et al., 2021), SIDD (Abdelhamed et al., 2018), and LoL (Wei et al., 2018). respectively.

## B  ADDITIONAL IMPLEMENTATION DETAILS

### B.1  INSTRUCTIONS FOR EACH TASK

In the following, we provide details into how we construct image and text instructions for each task.

**Semantic Segmentation.**   In this task, we transfer semantic labels to RGB images by binding each pixel with a unique color determined by its semantic class. We utilize the protocol released by Painter (Wang et al., 2022b) to define the semantic-color mapping. Instance-level prompt is adopted as the text instruction for this task. We derive the prompt following the template "semantic segmentation map of {caption}", where the caption is obtained by applying BLIP (Li et al., 2022a) to the query image.

**Depth Estimation.**   The depth map from NYUv2 is a one-channel image with the pixel value ranging from 0 to 10000. To obtain a RGB image, we scale the value of each pixel to $[0, 255]$ and then let R, G, B have the same re-scaled value. The text instruction for depth estimation is a task-level prompt: "depth map".

**Keypoint Detection.**   To convert keypoints into RGB images, we draw colored squares at the location of each keypoint. Each square occupies $9 \times 9$ pixels and its color is determined by the semantic category of that keypoint, and we adopt the same mapping strategy used for semantic segmentation. The text instruction for keypoint detection is a task-level prompt: "keypoint".

**Low-level Image Processing.**   We use the task-level prompt as the text instruction for three low-level image processing tasks. "a clean image without rain", "a sharp image without noise", and "a

bright image" are applied for image deraining, image denoising, and low-light image enhancement, respectively.

**Conditional Image Generation.** We adopt the same method used in visual understanding tasks to translate conditions to RGB images. Instance-level prompt, which is the caption of the output image obtained through BLIP, is used as the text instruction for these conditional image generation tasks.

### B.2 TRAINING AND INFERENCE DETAILS

We adopt a smooth L1 version of Eq. 2 to train our model. During training, we randomly drop 10% text-conditioning to improve classifier-free guidance sampling (Ho & Salimans, 2022). We load pre-trained weights from Stable Diffusion-v1.5-inpainting for fine-tuning. We randomly sample an input-output pair as the image instruction during training and adopt a fixed pair from the training dataset (same as Painter (Wang et al., 2022b)) as the image instruction at the inference time. By setting the random noise in the reverse diffusion process to 0 (i.e., a deterministic sampling), DDIM (Song et al., 2020a) manages to generate an image with fewer sampling steps compared to DDPM. We adopt DDIM sampling with 50 steps for inference.

## C ADDITIONAL RESULTS

**Additional comparison results.** Here we present additional comparison results on semantic segmentation (Figure 6), depth estimation (Figure 7), keypoint detection (Figure 8), low-light image enhancement (Figure 9), image deraining (Figure 10), image denoising (Figure 11), mask-to-image generation (Figure 12), depth-to-image generation (Figure 13), pose-to-image generation (Figure 14), and edge-to-image generation (Figure 15). These results further demonstrate the capability of `UniVis` to perform a large variety of computer vision tasks.

**Text-conditional image generation results.** We also explore an additional task: text-conditional image generation. This task can be fulfilled by directly applying `UniVis-sc` trained on four conditional image generation tasks where the query is set to a black image. We achieve a FID of 27.47 on the COCO validation set. This could be further improved by training a `UniVis-st` on this task. We also present some text-to-image generation results obtained from our method in Figure 16.

**Additional generalization results.** Here we showcase more generalization results from `UniVis` in Figure 17. `UniVis` exhibits promising performance when being applied to images from new domains/in-the-wild images and novel tasks with unseen conditions. This strong generalization capability, which is analogous to LLMs, again validates our design of building a universal solver for vision tasks. Gathering more diverse data for training UniVis could be promising to enhance generalizability and we plan to investigate this in future work.

## D OVERALL PERFORMANCE COMPARISON

For a more comprehensive performance comparison between Painter (Wang et al., 2022b), Prompt-Diffusion (Wang et al., 2023b), and `UniVis`, we include some discussions below. First, both Painter and PromptDiffusion experience a clear collapse or near breakdown on one of the three categories of vision tasks. For instance, on the conditional image generation tasks, Painter completely collapses while `UniVis` exhibits SOTA performance (see Table 3 and Figures 12, 13, 14, and 15). In other words, `UniVis` could handle at least one more category of vision tasks compared to its competitors. We conclude this in Table 7. For a more intuitive comparison, we draw a radar chart in Figure 5 to showcase the overall performance of different methods on three types of computer vision tasks. `UniVis` achieves the most balanced and comprehensive performance.

## E SCALING BEHAVIOR W.R.T. COMPUTING RESOURCES

Here we conduct an experiment where `UniVis` is trained using different amounts of computing resources. We present the results in Table 8. `UniVis` shows impressive scaling behavior where the performance improves with larger computing power.

Table 7: Capability of the proposed `UniVis` and its two direct competitors (Painter and PromptDiffusion) in handling three categories of vision tasks.

| Method | Visual Understanding | Low-level Image Processing | Conditional Image Generation |
|---|---|---|---|
| Painter | ✔ | ✔ | ✗ |
| PromptDiffusion | ✗ | ✔ | ✔ |
| UniVis | ✔ | ✔ | ✔ |

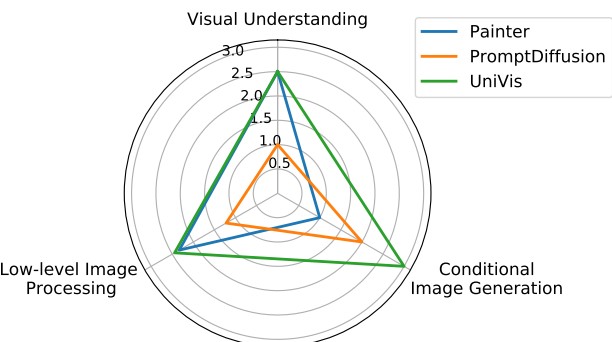

Figure 5: Overall performance comparison between `UniVis`, Painter, and PromptDiffusion on three categories of vision tasks. The score for each category is calculated based on the relative ranking among the three methods (e.g., 3 points for ranking 1st and 1 point for ranking 3rd) and averaged across tasks of each category.

## F  ABLATION STUDY FOR TASK PROMPTS

To further investigate the utility of task prompts, we perform two ablations for `UniVis-mc` and `UniVis-st` respectively. Results are provided in Table 9 and Table 10. As can be seen, task prompts are beneficial for some tasks (e.g., depth estimation and mask-to-image generation), but the gain by applying task prompts is very marginal for other tasks (e.g., deraining and denoising). Therefore, one can optionally apply task prompts during inference to strike a balance between better performance and extra human efforts.

## G  FEW-SHOT IN-CONTEXT INFERENCE

Here we extend `UniVis` to perform few-shot in-context inference. This is fulfilled by establishing the grid image with more input-output pairs. We report the results of `UniVis` under one-shot, two-shot, and four-shot settings in Table 11. We observe that there is a slight gain by introducing more visual instructions, and we think this could be further explored during training, which we leave to future work.

Table 8: Results of UniVis using different computing resources.

| Computing resources | Depth estimation | | | Denoising | | |
|---|---|---|---|---|---|---|
| | RMSE↓ | REL↓ | $\delta_1$↑ | PSNR↑ | SSIM↑ | LPIPS↓ |
| one 3090 GPU | 0.461 | 0.156 | 0.812 | 34.02 | 0.898 | 0.121 |
| four 3090 GPUs | 0.420 | 0.135 | 0.857 | 34.55 | 0.907 | 0.095 |
| eight A100 GPUs | 0.391 | 0.118 | 0.892 | 34.92 | 0.913 | 0.092 |

Table 9: Ablation study results of `UniVis-mc` regarding task prompts.

| Method | Depth estimation | | | Denoising | | | Mask-to-image |
|---|---|---|---|---|---|---|---|
| | RMSE↓ | REL↓ | $\delta_1$↑ | PSNR↑ | SSIM↑ | LPIPS↓ | FID↓ |
| `UniVis-mc` w/ task prompts | 0.421 | 0.131 | 0.863 | 34.58 | 0.909 | 0.095 | 30.4 |
| `UniVis-mc` w/o task prompts | 0.466 | 0.154 | 0.826 | 34.36 | 0.907 | 0.101 | 31.5 |

Table 10: Ablation study results of `UniVis-st` regarding task prompts.

| Method | Deraining | | |
|---|---|---|---|
| | PSNR↑ | SSIM↑ | LPIPS↓ |
| `UniVis-st` w/ task prompts | 22.62 | 0.598 | 0.302 |
| `UniVis-st` w/o task prompts | 22.60 | 0.595 | 0.306 |

Table 11: Results of `UniVis` using different shots of visual instructions.

| # of shots during inference | Depth estimation | | | Denoising | | |
|---|---|---|---|---|---|---|
| | RMSE↓ | REL↓ | $\delta_1$↑ | PSNR↑ | SSIM↑ | LPIPS↓ |
| one-shot | 0.420 | 0.135 | 0.857 | 34.55 | 0.907 | 0.095 |
| two-shot | 0.416 | 0.131 | 0.863 | 34.73 | 0.911 | 0.095 |
| four-shot | 0.413 | 0.130 | 0.863 | 34.75 | 0.911 | 0.095 |

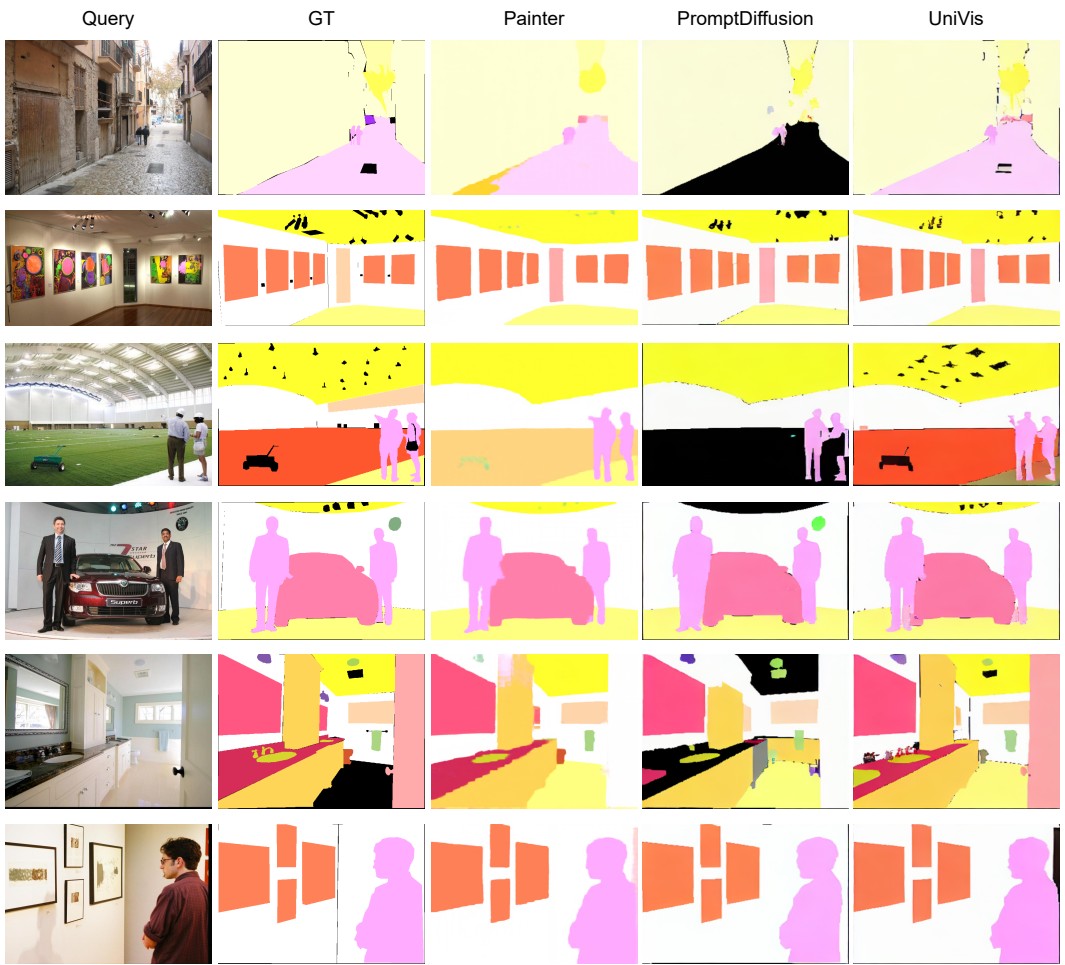

Figure 6: Visual comparison results on semantic segmentation.

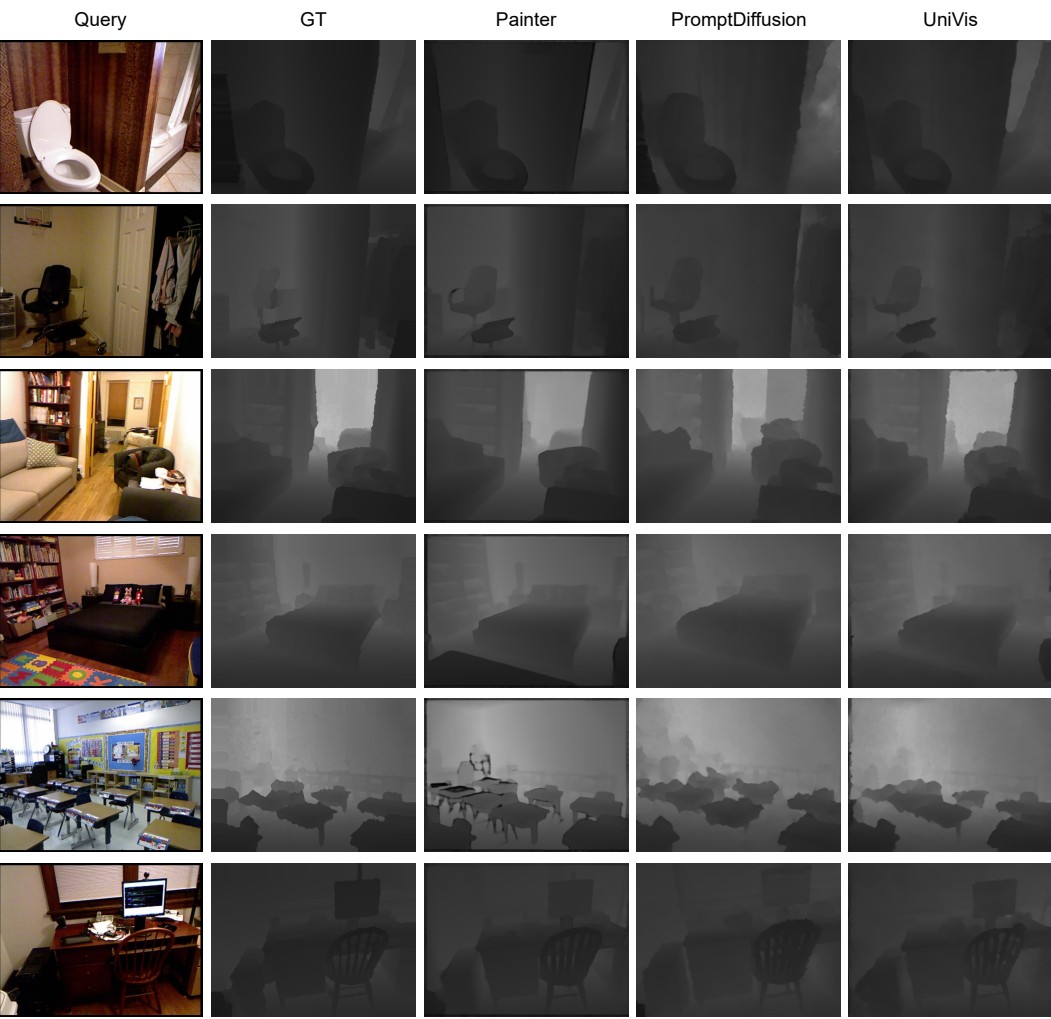

Figure 7: Visual comparison results on depth estimation.

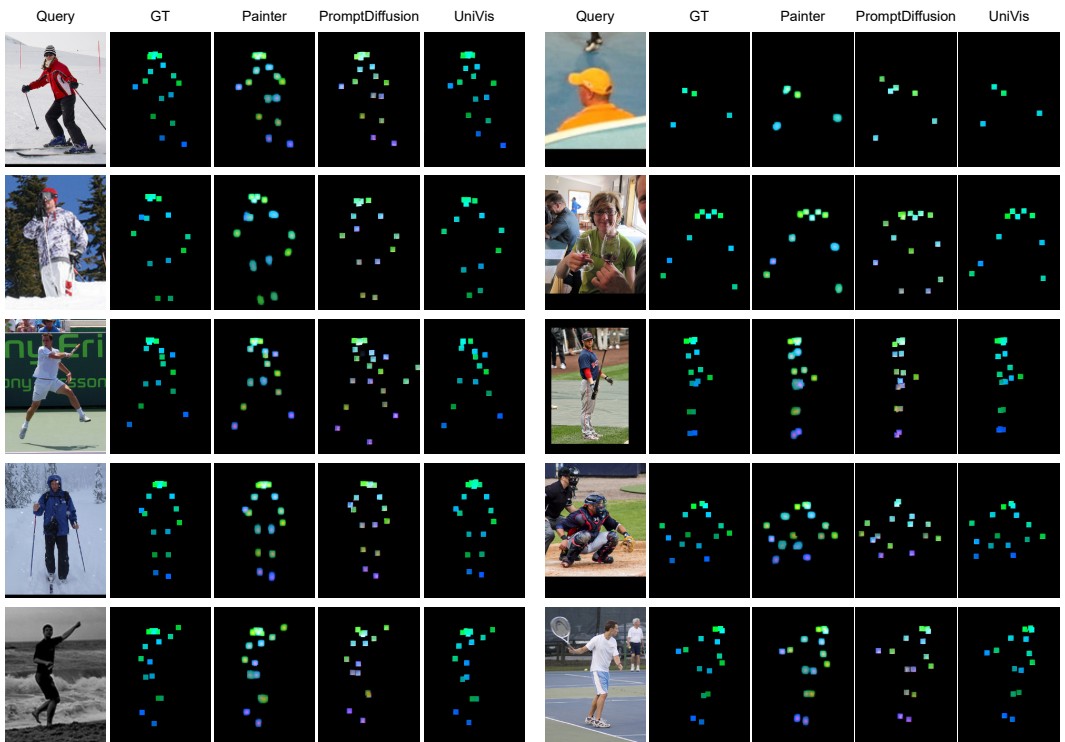

Figure 8: Visual comparison results on keypoint detection.

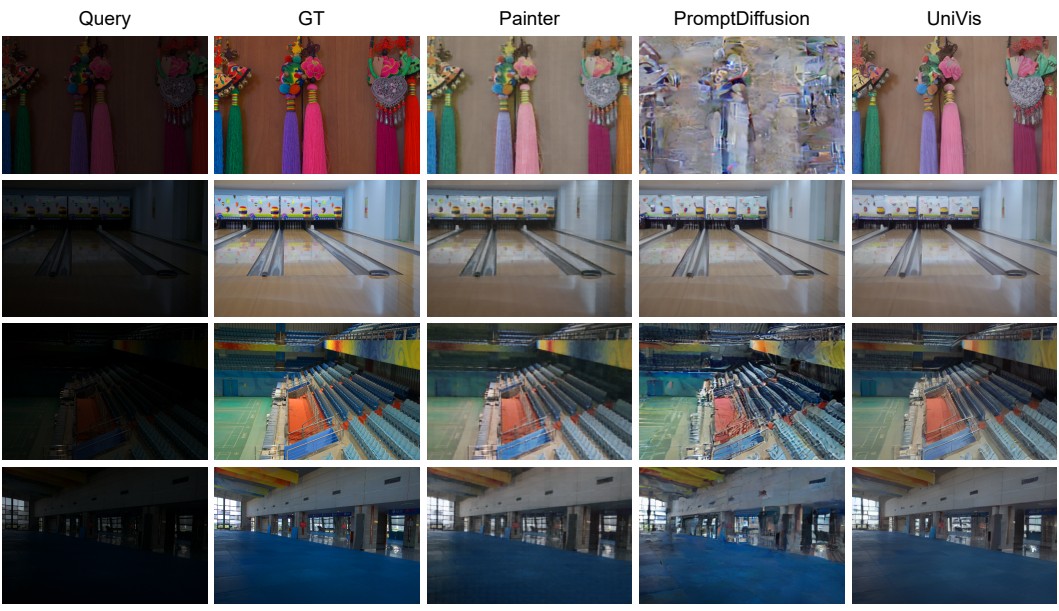

Figure 9: Visual comparison results on low-light image enhancement.

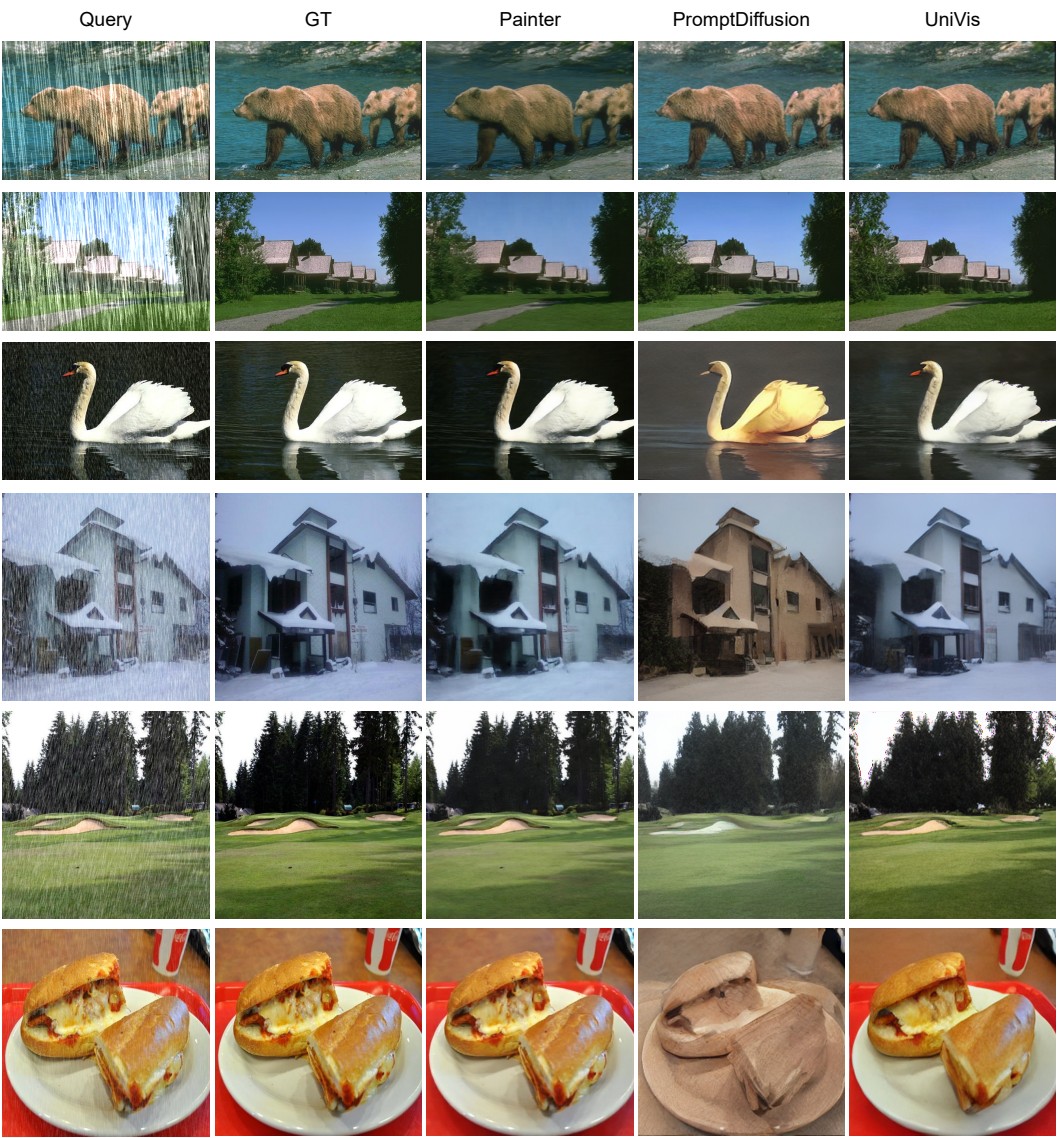

Figure 10: Visual comparison results on image deraining.

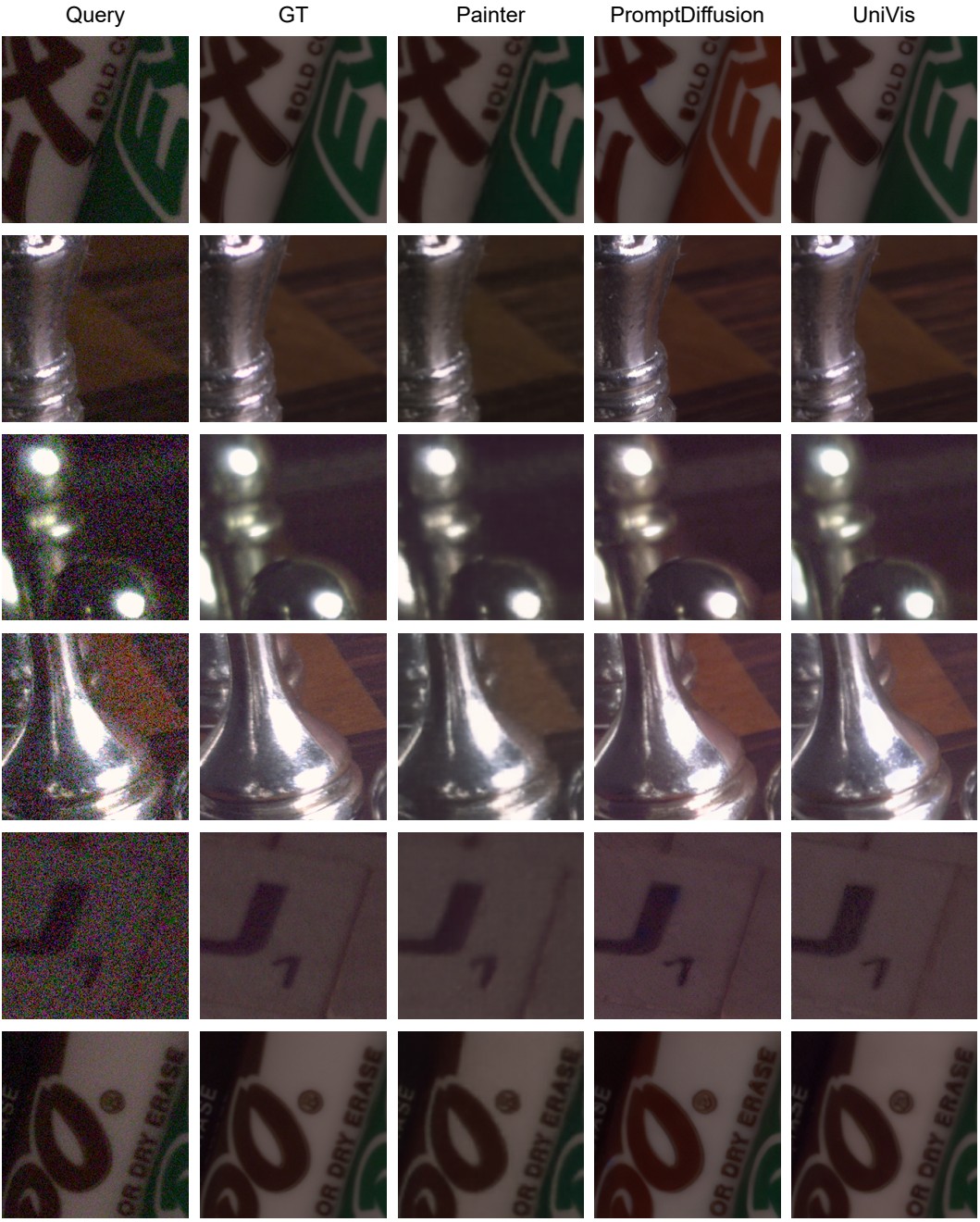

Figure 11: Visual comparison results on image denoising.

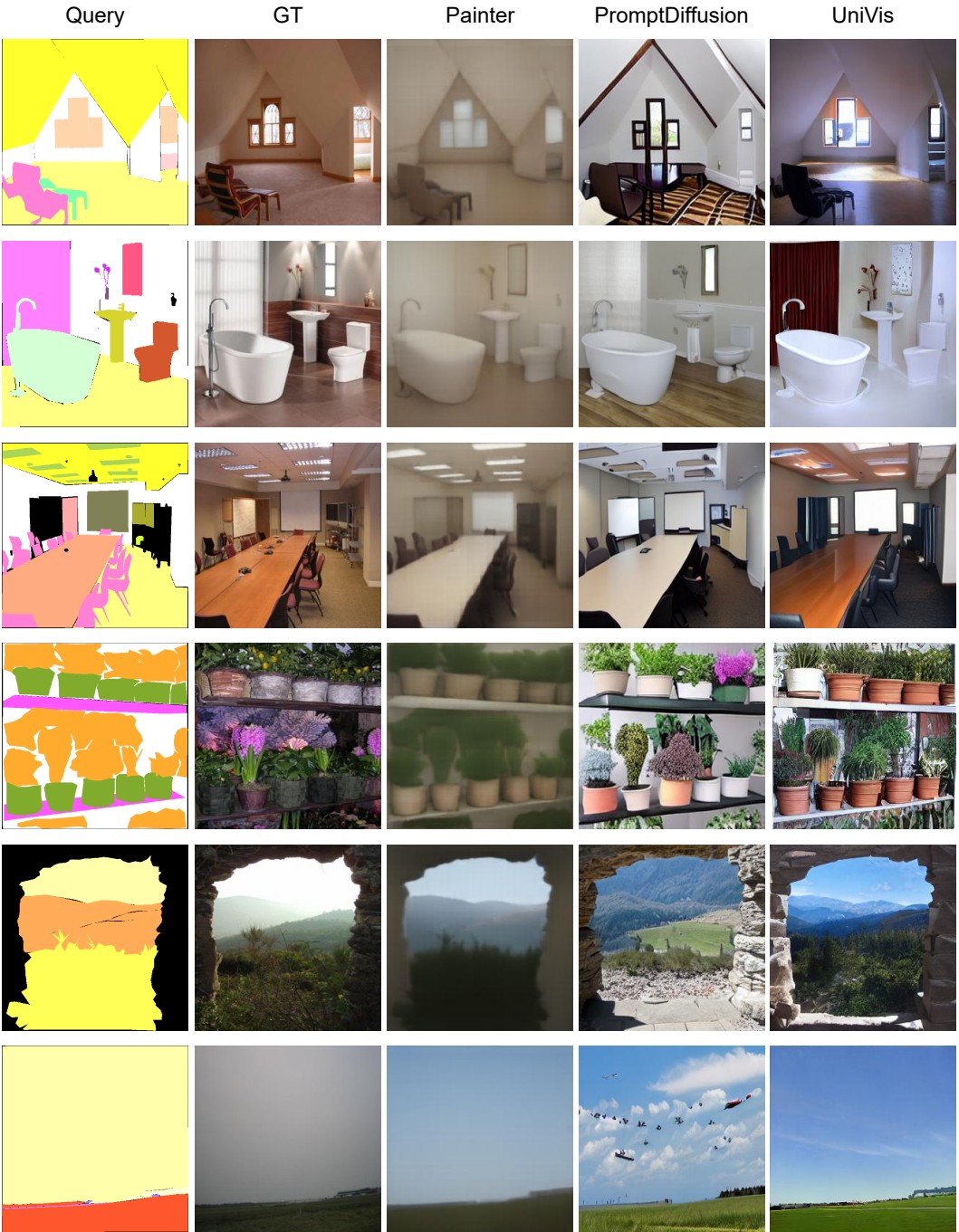

Figure 12: Visual comparison results on mask-to-image generation.

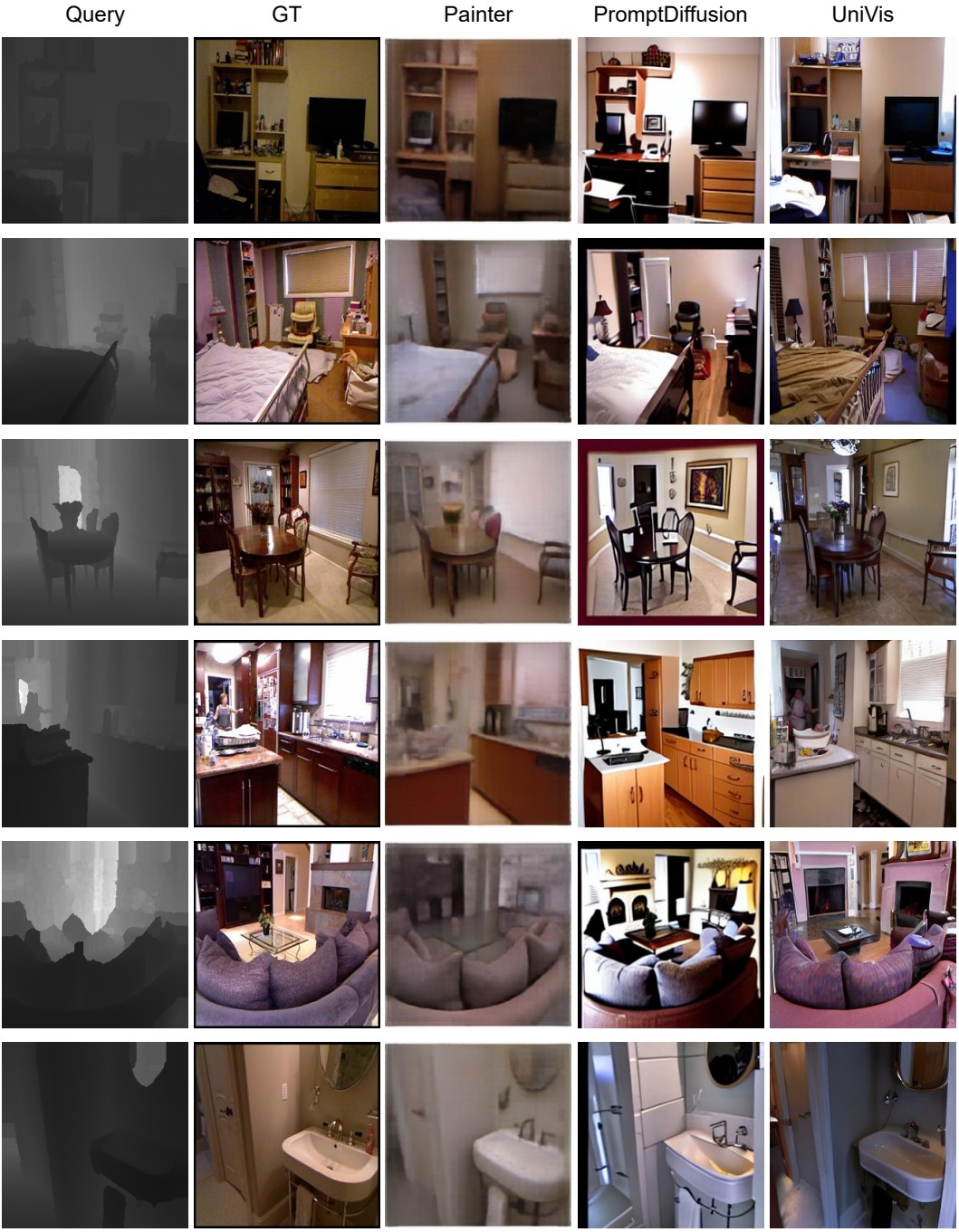

Figure 13: Visual comparison results on depth-to-image generation.

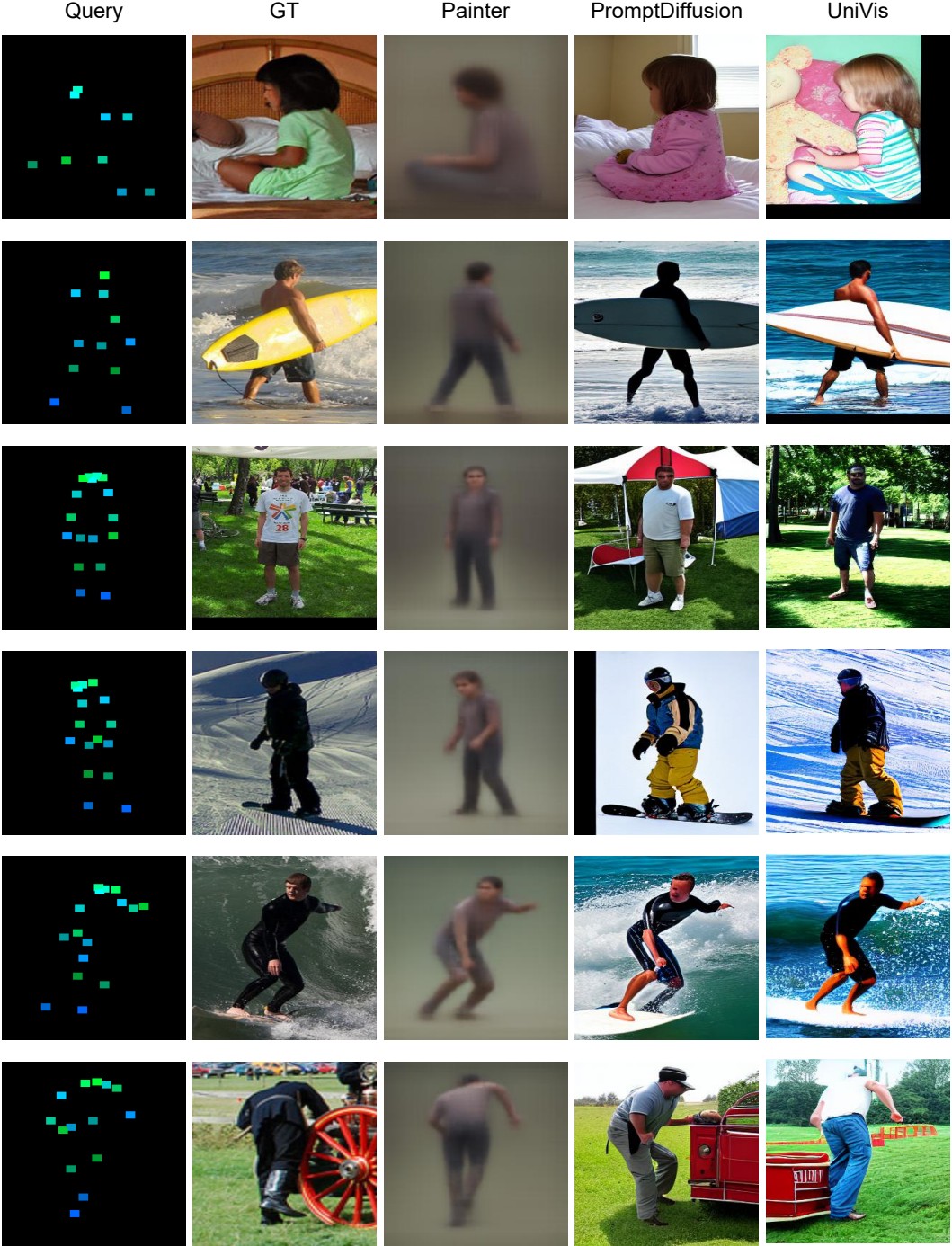

Figure 14: Visual comparison results on pose-to-image generation.

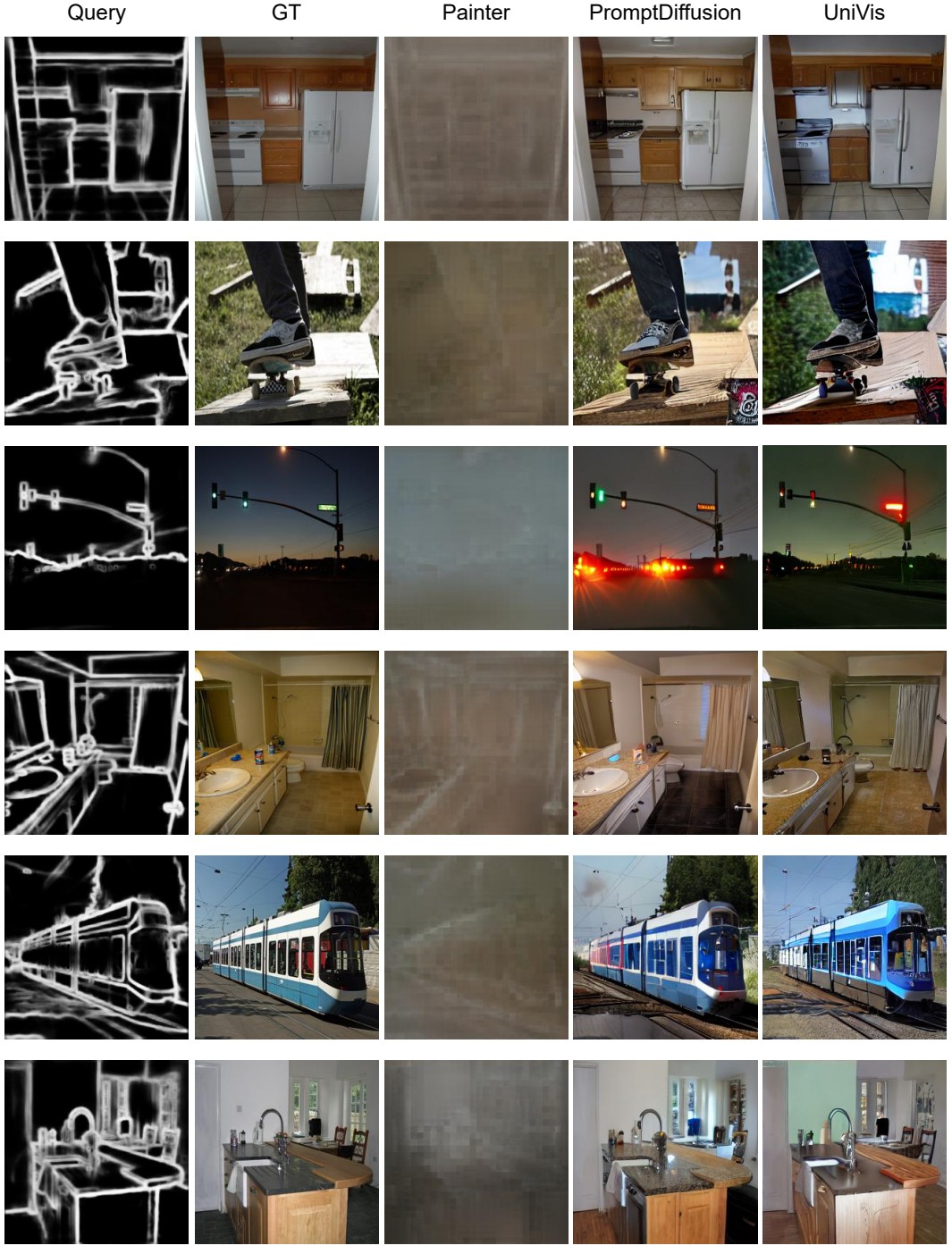

Figure 15: Visual comparison results on edge-to-image generation.

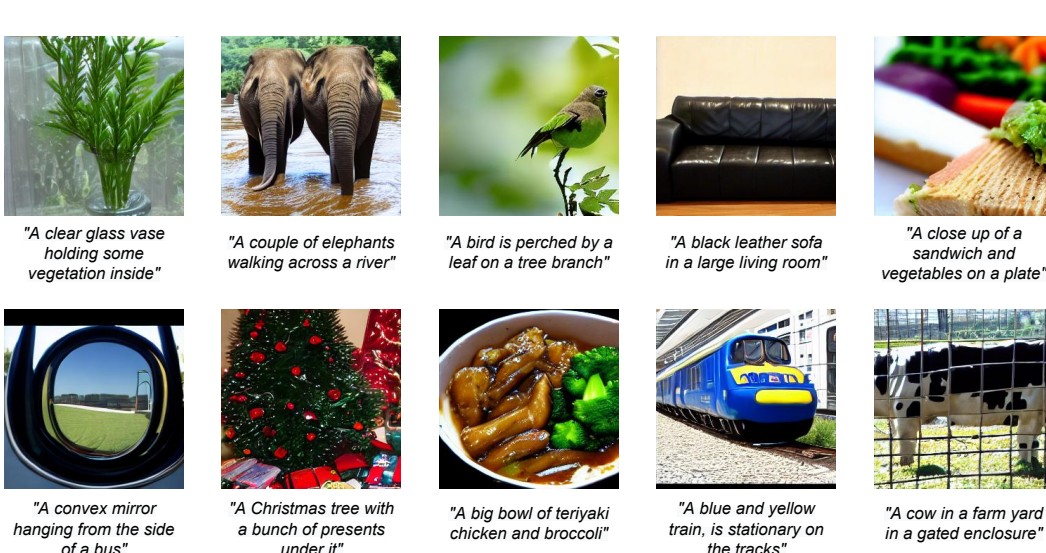

Figure 16: Text-to-image generation results of `UniVis`.

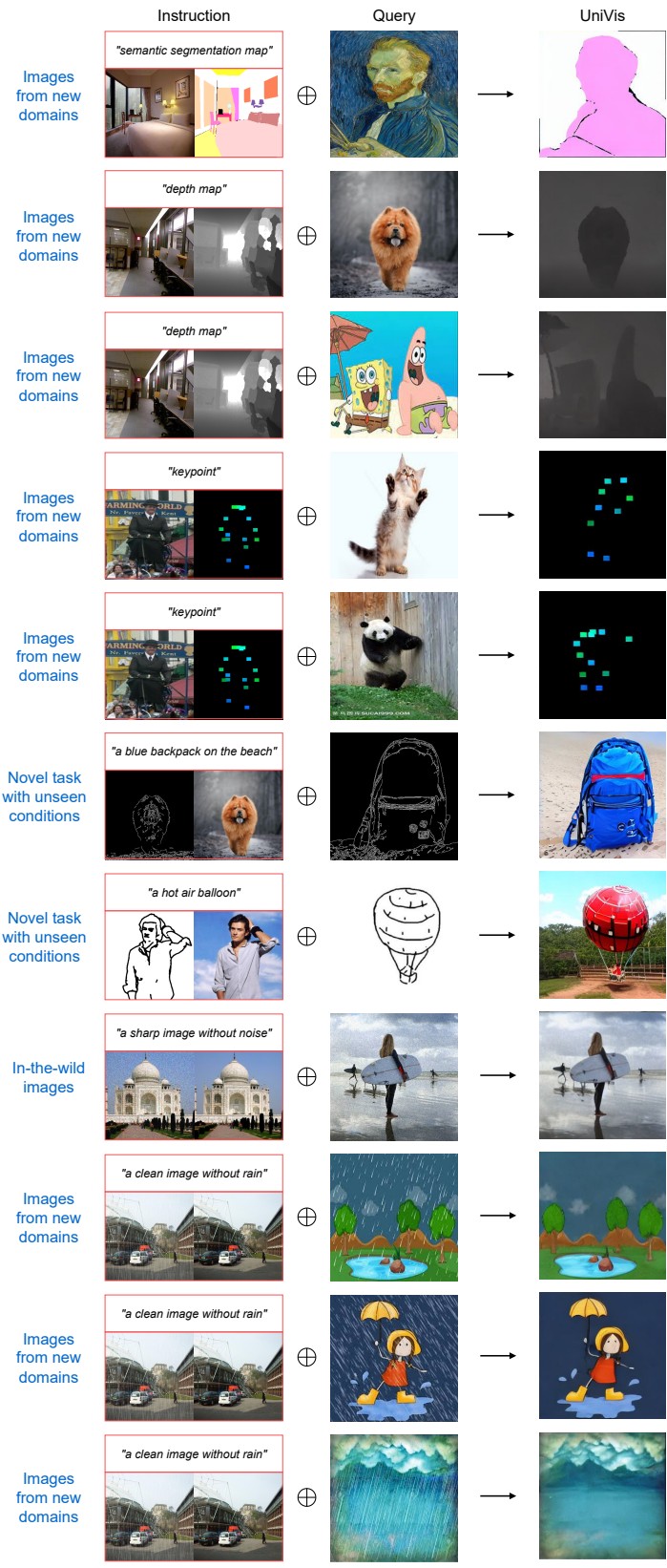

Figure 17: Additional generalization results from UniVis.

