# OpenReview forum: "UniVis: A Universal Framework for Computer Vision Tasks"
_ICLR.cc/2024/Conference — Submitted to ICLR 2024_

### Official Review · Reviewer_zYeV · 2023-10-31

**Soundness:** 2 fair
**Presentation:** 3 good
**Contribution:** 2 fair
**Rating:** 5
**Confidence:** 4

**Summary:**

This paper presents a framework called UniVis, with the aim of handling various categories and granularities of computer vision tasks. The framework is built upon a pre-trained stable diffusion model, thus enabling the transformation of some computer vision tasks into image generation (completion) tasks. The authors tested the framework on multiple benchmarks, and the experimental results showcased competitive performance.

**Strengths:**

The authors unify some computer vision tasks in terms of training format, viewing them as image generation tasks based on stable diffusion model (SD). With SD, this approach achieves better or competitive results on some benchmarks.

**Weaknesses:**

1. Constrained by the pretraining model (i.e., SD), UniVis performs better in image generation (Table 3) while average in visual understanding and low-level image processing (Table 1&2). This indicates that the framework is coupled with the training format of the pretraining model. Furthermore, the authors claim that there is no need for high training costs, but this method requires more parameters.

2. When using multiple datasets for joint training, there is no significant gain, even a slight decrease, for the three different categories of tasks (Table 4). This seems to suggest that this work is limited when using only one model to unify computer vision tasks (joint training doesn't yield gains). If it's just about unifying the training format, it doesn't seem to be very meaningful, as it doesn't outperform task-specific models on specific benchmarks.

3. Fig4 is significant as I personally believe one of the benefits of unifying computer vision tasks should be the ability to generalize to other unseen tasks (analogous to LLM), but unfortunately, only one figure is provided to demonstrate this. How to further enhance generalizability is an interesting point.

4. The testing scheme in the paper is similar to one-shot, how can it be extended to few-shot (>1)?

**Questions:**

See weakness.

---

> ### Author Response · Authors · 2023-11-20
> **Response to Reviewer zYeV (1/2)**
>
> We thank the reviewer for their valuable comments. We address your concerns below:
>
> **Q1: Average performance on visual understanding/low-level image processing and high training costs.**
> > Constrained by the pretraining model (i.e., SD), UniVis performs better in image generation (Table 3) while average in visual understanding and low-level image processing (Table 1&2). This indicates that the framework is coupled with the training format of the pretraining model. Furthermore, the authors claim that there is no need for high training costs, but this method requires more parameters.
>
> **A1:** (1) We would like to clarify that one major contribution of this paper is to build a  unified framework for vision tasks through **generative modeling** rather than commonly adopted contrastive learning or masked image modeling. Different from the well-known dictum by Richard Feynman, "What I cannot create, I do not understand", we find it very challenging for the generative models (like PromptDiffusion) to perform well on other vision tasks, especially the visual understanding tasks. Hence, the main aim of this work is making the generative model simultaneously work well on three distinct categories of vision tasks. To this end, we resort to a SOTA text-to-image model, which is one of the very few trained on web-scale data, and we carefully devise an instruction tuning pipeline to adapt it to downstream tasks. As you stated, we achieve competitive results on visual understanding and low-level image processing and exhibit SOTA performance on image generation (**see Figure 5 in the revised paper**). In contrast, our competing methods (Painter and PromptDiffusion) experience a clear collapse or near breakdown on one of the three categories of vision tasks (**see first table in general response #1**). Please refer to our general response #1 for more details.
>
> (2) We are a bit confused about the comment saying "requires more parameters". If you mean UniVis-st requires distinct model parameters to tackle different tasks (please let us know if you did not mean this), we would like to clarify that we also present models (UniVis-sc and UniVis-mc) that are jointly trained on multiple tasks (see Tables 2, 3, and 4), which use shared model parameters. The main contribution of our paper is a unified framework for visual task learning, and it can be employed to produce three types of models under different practice scenarios (e.g., the computing power at hand).
>
> **Q2: Joint training does not yield significant gains.**
> > When using multiple datasets for joint training, there is no significant gain, even a slight decrease, for the three different categories of tasks (Table 4). This seems to suggest that this work is limited when using only one model to unify computer vision tasks (joint training doesn't yield gains). If it's just about unifying the training format, it doesn't seem to be very meaningful, as it doesn't outperform task-specific models on specific benchmarks.
>
> **A2:** We would like to emphasize that our primary focus is to investigate how to induce a profound understanding of vision tasks (which involve very disparate visual signals and data domains) through a shared scheme of **generative modeling** instead of seeking gains with multi-task training. Please see our general response #2 for more details.
>
> **Q3: More generalization results.**
> > Fig4 is significant as I personally believe one of the benefits of unifying computer vision tasks should be the ability to generalize to other unseen tasks (analogous to LLM), but unfortunately, only one figure is provided to demonstrate this. How to further enhance generalizability is an interesting point.
>
> **A3:** We have added more results in the updated paper (**please refer to Figure 17**). They verify the generalization capability of the proposed framework. We totally agree that performing out-of-distribution inference is one of the intriguing properties of unified models. Gathering more diverse data for training UniVis could be promising to enhance generalizability and we plan to investigate this in future work.

---

> > ### Author Response · Authors · 2023-11-20
> > **Response to Reviewer zYeV (2/2)**
> >
> > **Q4: Few-shot testing scheme.**
> > > The testing scheme in the paper is similar to one-shot, how can it be extended to few-shot (>1)?
> >
> > **A4:** Few-shot in-context inference can be conducted by establishing the grid image with more input-output pairs. Following this, we made some attempts under two-shot and four-shot settings during inference and reported the results below (also added to Table 11 in the revised paper). We observe that there is a slight gain by introducing more visual instructions, and we think this could be further explored during training, which we leave to future work.
> >
> > |                                 |   |                  | Depth estimation |                        |   |                |    Denoising   |                   |
> > |---------------------------------|:-:|:----------------:|:----------------:|:----------------------:|:-:|:--------------:|:--------------:|:-----------------:|
> > | **# of shots during inference** |   | RMSE$\downarrow$ |  REL$\downarrow$ | $\delta_{1}$$\uparrow$ |   | PSNR$\uparrow$ | SSIM$\uparrow$ | LPIPS$\downarrow$ |
> > | one-shot                        |   |       0.420      |       0.135      |          0.857         |   |      34.55     |      0.907     |       0.095       |
> > | two-shot                        |   |       0.416      |       0.131      |          0.863         |   |      34.73     |      0.911     |       0.095       |
> > | four-shot                       |   |       0.413      |       0.130      |          0.863         |   |      34.75     |      0.911     |       0.095       |

---

> > > ### Author Response · Authors · 2023-11-22
> > > **Official Comment by Authors**
> > >
> > > Dear Reviewer zYeV,
> > >
> > > We sincerely appreciate your valuable feedback on our submission. As the author-reviewer discussion nears its end, we are eager to know if our responses have addressed your concerns. We are also more than willing to engage in further discussions if needed.
> > >
> > > Thank you again for your time and effort!

---

> > > > ### Comment · Reviewer_zYeV · 2023-11-22
> > > >
> > > > Thanks for the authors’ efforts in rebuttal.
> > > > However, after reviewing the responses and considering the feedback from other reviewers, I have chosen to maintain my original score. My primary concern is that the proposed unified framework does not display significant improvements across the various tasks, nor does it exhibit any emergent properties. While the authors emphasize their focus on exploring methods to induce a deeper understanding of vision tasks, the lack of noticeable improvements or new capabilities (just some figures are shown) makes the unification less meaningful, as I previously mentioned. Additionally, although the authors highlight that the framework does not rely on SD (and masked data modeling would be sufficient), all conducted experiments are exclusively based on SD.

---

> > > > > ### Author Response · Authors · 2023-11-22
> > > > > **Clarification by Authors**
> > > > >
> > > > > Thanks for your feedback. We would like to provide some clarifications, which we believe can address your main concerns. We are open to any further discussion and would appreciate a reassessment of our work.
> > > > >
> > > > > **1. Lack of noticeable improvements & less meaningful unification.**
> > > > >
> > > > > As shown in the general response, the two direct competitors Painter and PromptDiffusion experience a **CLEAR COLLAPSE** or **NEAR BREAKDOWN** on one of the three categories of vision tasks. In contrast, the proposed UniVis can **WORK WELL** on **ALL** the classes of vision tasks. Furthermore, UniVis achieves **SOTA** results in many of generation tasks even compared to specialized methods (ControlNet), with 11% / 73% lower FID than PromptDiffusion / Painter. Hence, we believe this is a **noticeable improvement** from the perspective of unifying discrimination and generation.
> > > > >
> > > > > **2. Does not exhibit any emergent properties.**
> > > > >
> > > > > Following the published PromptDiffusion paper, we opted to visualize the generated images to evaluate the generalization capability, as it seems non-trivial to report the quantitative results. The results in Figure 4 and the additional ones in Figure 17 clearly show that UniVis generates realistic images from **UNSEEN CONDITIONS** during training, which could be viewed as an emergent property.
> > > > >
> > > > > **3. "The authors highlight that the framework does not rely on SD."**
> > > > >
> > > > > **WE DID NOT HAVE THIS CLAIM.** Instead, we think the reliance on SD **benefits** the unifying ability due to the generative and reasoning power of the pre-trained SD (as we stated in the abstract and introduction). Extensive results demonstrate the ability of UniVis in **simultaneously handling discrimination and generation**, which is beyond the reach of prior methods.

---

### Official Review · Reviewer_ZgiD · 2023-11-02

**Soundness:** 2 fair
**Presentation:** 3 good
**Contribution:** 2 fair
**Rating:** 5
**Confidence:** 4

**Summary:**

The paper introduces UniVis, a universal learning framework designed for various computer vision tasks. Drawing inspiration from the success of large language models (LLMs) in natural language processing (NLP), UniVis seeks to offer a unified solution for visual tasks. Based on the text-to-image diffusion model, Stable Diffusion (SD), the framework leverages instruction tuning to adapt pre-trained knowledge to diverse downstream vision tasks. This approach employs an image completion framework, where input comprises a query image paired with another input-output image related to the target task. Through this, the model discerns the desired output for the query. The central tenets include:  1. Vision tasks can be represented as unique input-output pairs. 2. Vision tasks can benefit from optional text prompts. 3. The reasoning ability of SD can be harnessed for diverse vision tasks.
The authors undertook comprehensive experiments across ten vision tasks and three distinct training methodologies, aiming to ignite further exploration into fostering a deeper comprehension of vision tasks via a unified generative modeling approach.

**Strengths:**

[Task] The undertaking of employing SD as an interface for diverse downstream tasks presented in this study is intriguing. It enables the pre-trained knowledge to be adaptable and applicable across various downstream vision tasks.

[Experimental Results] The authors carried out a thorough evaluation across many downstream tasks.

[Paper Writing] The manuscript is well written, effectively conveying the primary concepts.

**Weaknesses:**

[Model Performance] While the authors claim reduced computational resource usage, the model's performance significantly lags behind the open-sourced baseline model, Painter [1], as evident in Table 2.

[Universality with Text Instructions] The framework's universality is some kind of diminished by its reliance on task-specific text instructions. A more compelling setup would operate without any task prompts. Text instructions should be supplemental, enriching tasks like text-to-image generation with finer details, rather than being a mandatory prerequisite for all tasks.

[Task Prompt Limitation] The inclusion of task prompts detracts from the intriguing properties of in-context instruction tuning. I think, could be wrong, the essence of in-context learning lies in the model's emergent properties, deciphering the logic and connections within paired VISUAL inputs to undertake related tasks. Ideally, we'd want to furnish only in-context visual cues, enabling the model to manage a myriad of downstream tasks. Yet, the current design seems to veer away from this ideal.

[1] Wang, Xinlong, Wen Wang, Yue Cao, Chunhua Shen, and Tiejun Huang. "Images speak in images: A generalist painter for in-context visual learning." In Proceedings of the IEEE/CVF Conference on Computer Vision and Pattern Recognition, pp. 6830-6839. 2023.

**Questions:**

Most of my questions are in the weakness section. In addition, I'm curious how the model performs without relying on task prompts. Have you conducted any ablation studies to shed light on this aspect?

---

> ### Author Response · Authors · 2023-11-20
> **Response to Reviewer ZgiD**
>
> We thank the reviewer for their insightful comments. We are willing to address all the concerns raised in your review:
>
> **Q1: Model performance.**
> > While the authors claim reduced computational resource usage, the model's performance significantly lags behind the open-sourced baseline model, Painter, as evident in Table 2.
>
> **A1:** Please see our general response #1.
>
> **Q2: Universality with text instructions.**
> > The framework's universality is some kind of diminished by its reliance on task-specific text instructions. A more compelling setup would operate without any task prompts. Text instructions should be supplemental, enriching tasks like text-to-image generation with finer details, rather than being a mandatory prerequisite for all tasks.
>
> **A2:** (1) Text instructions are indeed supplemental in our design. An ablation result regarding different types of text instructions is given in Table 5. To further explore this, we conducted more experiments during rebuttal and provided the results below (added to Tables 9 and 10 in the revised paper as well).
>
> |               |   |              | Depth estimation |                   |   |              |    Denoising   |             |   |  Mask-to-image  |
> |----------------------------|---|:----------------:|:----------------:|:----------------------:|:-:|:--------------:|:--------------:|:-----------------:|:-:|:---------------:|
> | **Method**                 |   | RMSE$\downarrow$ |  REL$\downarrow$ | $\delta_{1}$$\uparrow$ |   | PSNR$\uparrow$ | SSIM$\uparrow$ | LPIPS$\downarrow$ |   | FID$\downarrow$ |
> | UniVis-mc w/ task prompts  |   |       0.421      |       0.131      |          0.863         |   |      34.58     |      0.909     |       0.095       |   |       30.4      |
> | UniVis-mc w/o task prompts |   |       0.466      |       0.154      |          0.826         |   |      34.36     |      0.907     |       0.101       |   |       31.5      |
>
> |                            |   |                |    Deraining   |                   |
> |----------------------------|---|:--------------:|:--------------:|:-----------------:|
> | **Method**                 |   | PSNR$\uparrow$ | SSIM$\uparrow$ | LPIPS$\downarrow$ |
> | UniVis-st w/ task prompts  |   |      22.62     |      0.598     |       0.302       |
> | UniVis-st w/o task prompts |   |      22.60     |      0.595     |       0.306       |
>
> As can be seen, text instructions are beneficial for some tasks (e.g., semantic segmentation, depth estimation, and mask-to-image), but the gain by applying task prompts is very marginal for other tasks (e.g., deraining and denoising). Therefore, one can optionally apply text instructions during inference to strike a balance between better performance and extra human efforts. (2) We do not think incorporating task-specific text instructions undermines the universality of the proposed framework. From the perspective of a user, one is aware of the task name when constructing the image instruction (input-output pair), thus the text instruction can be given by 1) an empty string, or 2) a task-level prompt (e.g., "depth map"), or 3) an instance-level prompt (could use the help from some off-the-shelf tools such as image captioning methods). Our claimed "universality" lies in the shared framework for various computer vision tasks.
>
> **Q3: Task prompt limitation.**
> > The inclusion of task prompts detracts from the intriguing properties of in-context instruction tuning. I think, could be wrong, the essence of in-context learning lies in the model's emergent properties, deciphering the logic and connections within paired VISUAL inputs to undertake related tasks. Ideally, we'd want to furnish only in-context visual cues, enabling the model to manage a myriad of downstream tasks. Yet, the current design seems to veer away from this ideal.
>
> **A3:** We agree that the essence of in-context learning may lie in the model's emergent properties. The model we utilize is Stable Diffusion (SD) which translates textual inputs into realistic images, and a bunch of prior works has demonstrated that the knowledge learned in SD contains a rich understanding of both visual and linguistic signals. In light of this, we opt to exploit the inclusion of task prompts for UniVis, which we believe could be fruitful for unlocking the unifying capability of a pre-trained SD model.
>
> **Q4: Ablation studies regarding task prompts.**
> > I'm curious how the model performs without relying on task prompts. Have you conducted any ablation studies to shed light on this aspect?
>
> **A4:** An ablation result regarding different types of text instructions is given in Table 5. We conducted two more experiments during rebuttal and provided the results in Tables 9 and 10 in the revised paper. As can be seen, text instructions are beneficial for some tasks (e.g., semantic segmentation, depth estimation, and mask-to-image), but the gain by applying task prompts is very marginal for other tasks (e.g., deraining and denoising).

---

> > ### Author Response · Authors · 2023-11-22
> > **Official Comment by Authors**
> >
> > Dear Reviewer ZgiD,
> >
> > We sincerely appreciate your valuable feedback on our submission. As the author-reviewer discussion nears its end, we are eager to know if our responses have addressed your concerns. We are also more than willing to engage in further discussions if needed.
> >
> > Thank you again for your time and effort!

---

> > > ### Comment · Reviewer_ZgiD · 2023-11-22
> > >
> > > Dear authors, thank you for your time in addressing my comments. However, my key concern is still on the [Task Prompt Limitation]: The inclusion of task prompts detracts from the intriguing properties of in-context instruction tuning. I think, could be wrong, the essence of in-context learning lies in the model's emergent properties, deciphering the logic and connections within paired VISUAL inputs to undertake related tasks. Ideally, we'd want to furnish only in-context visual cues, enabling the model to manage a myriad of downstream tasks. Yet, the current design seems to veer away from this ideal.
> > >
> > > Therefore, I will keep my rating as borderline reject. I think this direction is interesting, but the absence of emerging properties makes this work less exciting. I encourage the authors to keep exploring this direction!

---

> > > > ### Author Response · Authors · 2023-11-23
> > > > **Clarification by Authors**
> > > >
> > > > Thanks for your feedback. We want to make some clarifications to avoid any possible misunderstandings.
> > > >
> > > > **1. Task prompt limitation.**
> > > >
> > > > Could you please provide some elaborations on why our early response did not address your concerns regarding the "Task Prompt Limitation"? The pre-trained SD is a text-to-image generation model, and its generative and reasoning power is exhibited in **the connection between images and language.** Therefore, we want to furnish both visual and linguistic cues to better unlock the in-context inference ability of SD.
> > > >
> > > > **2. Absence of emerging properties.**
> > > >
> > > > As clearly shown in Figure 4(a) and the 6th and 7th rows in Figure 17, UniVis **DOES** generate realistic images from **UNSEEN CONDITIONS** during training, which could be viewed as an emergent property of UniVis.
> > > >
> > > > We are open to any further discussion and would appreciate a reassessment of our work. Thank you.

---

### Official Review · Reviewer_YRyF · 2023-11-04

**Soundness:** 4 excellent
**Presentation:** 4 excellent
**Contribution:** 3 good
**Rating:** 8
**Confidence:** 3

**Summary:**

The authors present and evaluate "UniVis," which is an approach to training and obtaining inferences from a Stable Diffusion model across a variety of tasks from three distinct categories of tasks. The model is trained using an instruction image pair that demonstrates the task to be perform (e.g., for the depth estimation task, the "instruction" or "example" pair would be an RGB query image of a scene and a corresponding depth image as example output). At training time, the model is also presented an input query image of the kind expected for the task (e.g. a RGB scene) and trained to produce the ground-truth output for that particular example. This can be supplemented with a textual prompt to further condition the denoising U-Net within the Stable Diffusion model (e.g. the text instruction for the depth estimation task would be "depth map").

This is essentially an instruction-tuning framework for Stable Diffusion, where the instruction is an image pair that demonstrates the task, optionally supplemented with a text description of the task.

Most of the evaluations are conducted by retraining UniVis for a specific task (e.g. depth estimation, or denoising, or pose-to-image image generation). Because of this, this paper is largely a demonstration of the generalizability of the training _process_, rather than the generalizability of a single trained model across the multiple kinds of tasks.

However, one experiment is conducted to show that a single trained model can also generalize across tasks from the three task categories (from image understanding: depth estimation; from low-level image processing: denoising; and from conditional image generation: mask-to-image). One experiment also demonstrates that a single trained model can generalize across tasks from within one category: a model was trained to be able to perform inference for four conditional image generation tasks (mask-to-image, depth-to-image, pose-to-image, and edge-to-image).

The results appear to show performance from the single-task UniVis on par with the "Painter" model of Wang et al. (https://arxiv.org/abs/2212.02499), when Painter's training is constrained to use the same amount of computing power as UniVis's authors used for UniVis.

Results did not appear to significantly deteriorate when UniVis was trained as a multi-task (but still single-category) model for the four conditional image generation tasks (see Table 3). Nor did results appear to significantly deteriorate when UniVis was trained on multiple tasks that spanned the three categories of tasks (see Table 4).

**Strengths:**

The ability to produce a single trained model that can generalize across diverse computer vision tasks by simple altering the "instructions" would be very useful. While instruction tuning is not original, the specific construction of the "instructions" for UniVis _is_ original. It has much in common with the instruction pairs from "Painter" (https://arxiv.org/abs/2212.02499), but instead of masking random subregions of the target images, the authors here train UniVis by masking the entire target image and train UniVis to generate the complete output. It did not seem to me that this would be significant, so I was quite surprised that it appears this change in masking strategy is crucial to unlocking the unifying capability of a pre-trained Stable Diffusion model (see Table 5).

**Weaknesses:**

**I note that these concerns have largely been addressed in response and revision, but I leave these comments here for context. I have updated my scoring, however.**

I will write this in the first person, to directly address the authors, on the hope that these comments may help improve the paper.

My biggest concern is that the results in the tables cannot be understood without knowing the variation that might be produced from one repetition of an experiment to the next. It is crucial that you state whether you only ran each experiment once, or whether the numbers you are reporting are the averages across several trials. If the latter, it is also essential to provide some measure of variation (standard deviation, confidence interval). If you have no estimation of the experiment-to-experiment variation, how do you know that what you are observing as differences in the tables is not simply noise?

In two places, you make assertions / claims that could use further elaboration or specificity. They are also maybe unnecessary in light of what you are actually showing in this paper.

1. At the bottom of p. 1, you say that LLMs exhibit "superior linguistic understanding due to the highly organized and structured nature of language." But images also have significant structure. You go on to say that the "disparity between low-level features and high-level semantics... is typically more pronounced in vision than in text." This is all very vague. Do you need to say it? If you do, could you be more precise about how you are assessing "disparity" or the degree of organization or structure?
2. At p. 6, you say that "patch-level inpainting that resembles word completion training schemes in NLP is not adequate for a holistic and profound understanding of vision tasks because images _have much lower semantic density than texts_." This is vague. I think I know what you mean: that there is a lot of redundancy and spatial correlation in images that is not present in text. Could you make this more precise?

Some of the concepts are under-explained, or used without any explanation:

1. DDIM is mentioned at p. 16 (B.2) without any explanation.
2. I know U-Net is a well-understood term-of-art by now, but I think it could still use a brief explanation of its purpose, given that it is what is being trained to fit the distribution of latent codes.
3. You use the phrase "spatial-wise concatenation" at p. 4 (first paragraph of 3.1). Can you describe this? I think you simply mean you can stitch the images together in a grid as you visualize in Figure 2, right?

Some of the phrasing is unclear or awkward. I can provide some suggestions for improvement.

1. At p. 3, you talk about "three _implementations_" of UniVis. But I would hesitate to call these different _implementations_. I think the contribution of your paper is that these are all the _same_ implementation, but simply trained in three different regimes (single-task, single-category, and multi-category). I prefer the phrase you use later: "three types of model training."
2. In the first sentence of the abstract: the word "tam" seems to be a typo.
3. In the introduction, "all-rounder" is unclear.
4. In the introduction, the sentence "The Challenges are in three aspects, elaborated in the following" is an awkward sentence. I suggest simply: "There are three main challenges."

**Questions:**

1. Why do you ignore the apparently better-performing comparator models in several of the tables when reporting the "best" and "second best"? For instance, in Table 1, why does OneFormer not get bolded as the "best"? If you are ignoring specialized models in your ranking of best and second best, you should include this caveat in your description of "best."

2. I see that you trained a single-category UniVis on the four conditional image generation tasks. Did you attempt training your multi-task/single-category UniVis on either of the two other categories? If not, why did you choose conditional image generation as the category to try as the single-category UniVis?

3. I see that for multi-category UniVis, you selected depth estimation, denoising, and mask-to-image. Did you attempt training other combinations of tasks across the categories (e.g. semantic segmetation + deraining + pose-to-image)? If not, why did you choose the three you chose?

4. Do you expect some combinations of tasks to be particularly difficult for UniVis to be trained for at the same time?

5. If you have no estimation of the run-to-run variation, how do you know that what you are observing as differences in the tables is not simply noise?

---

> ### Author Response · Authors · 2023-11-20
> **Response to Reviewer YRyF (1/2)**
>
> Thank you for the detailed and constructive review. We are happy to follow your comments to further improve this paper:
>
> **Q1: Estimation of the experiment-to-experiment variation**
> > My biggest concern is that the results in the tables cannot be understood without knowing the variation that might be produced from one repetition of an experiment to the next. It is crucial that you state whether you only ran each experiment once, or whether the numbers you are reporting are the averages across several trials. If the latter, it is also essential to provide some measure of variation (standard deviation, confidence interval). If you have no estimation of the experiment-to-experiment variation, how do you know that what you are observing as differences in the tables is not simply noise?.
>
> **A1:** We have updated some tables in the paper to include the average scores and standard deviations across three trials. From the added results, we could find that the proposed UniVis is very stable with small stds. Due to limited time and computing resources, we are unable to run all experiments multiple times during rebuttal. We promise to update all tables in the final version if this paper is accepted.
>
> **Q2-1: Vague assertions/claims in the introduction.**
> > At the bottom of p. 1, you say that LLMs exhibit "superior linguistic understanding due to the highly organized and structured nature of language." But images also have significant structure. You go on to say that the "disparity between low-level features and high-level semantics... is typically more pronounced in vision than in text." This is all very vague. Do you need to say it? If you do, could you be more precise about how you are assessing "disparity" or the degree of organization or structure?
>
> **A2-1:** In the second paragraph of Section 1, we have changed "...due to the highly organized and structured nature of language." into "...due to the semantic-dense nature of language created by humans." We find that the description regarding "disparity" is indeed vague, and we thus remove it to avoid misunderstandings.
>
> **Q2-2: Vague assertions/claims in Section 3.2.**
> > At p. 6, you say that "patch-level inpainting that resembles word completion training schemes in NLP is not adequate for a holistic and profound understanding of vision tasks because images have much lower semantic density than texts." This is vague. I think I know what you mean: that there is a lot of redundancy and spatial correlation in images that is not present in text. Could you make this more precise?
>
> **A2-2:** We have updated the description as follows (see last paragraph of Section 3.2): "...due to the fact that the correlation between pixels is much stronger than that between words (e.g., this redundancy presented in images makes the model readily inpaint a patch with neighboring patches)."
>
> **Q3-1: Under-explained concepts regarding DDIM.**
> > DDIM is mentioned at p. 16 (B.2) without any explanation.
>
> **A3-1:** We have added a brief introduction to DDIM in the revised paper as follows (see Appendix B.2): "By setting the random noise in the reverse diffusion process to 0 (i.e., a  deterministic sampling), DDIM manages to generate an image with fewer sampling steps compared to DDPM."
>
> **Q3-2: Under-explained concepts regarding U-Net.**
> > I know U-Net is a well-understood term-of-art by now, but I think it could still use a brief explanation of its purpose, given that it is what is being trained to fit the distribution of latent codes.
>
> **A3-2:** We have added a brief explanation of the purpose of U-Net in the revised paper as follows (see first paragraph of Section 3.2): "Specifically, a denoising U-Net is trained to fit the distribution of latent codes, which models the reverse diffusion process. Taking the noisy latent and the time step as input, this U-Net is further conditioned on the textual embeddings extracted through a text encoder CLIP via cross-attention to produce the output at the current time step."
>
> **Q3-3: Under-explained concepts regarding spatial-wise concatenation.**
> > You use the phrase "spatial-wise concatenation" at p. 4 (first paragraph of 3.1). Can you describe this? I think you simply mean you can stitch the images together in a grid as you visualize in Figure 2, right?
>
> **A3-3:** Yes it is, we have appended a brief explanation of "spatial-wise concatenation" in the revised paper as follows (see first paragraph of Section 3.1): "we can implement spatial-wise concatenation of any input, out, and query samples (i.e., stitching all the images together into a grid as illustrated in Figure 2)."

---

> > ### Author Response · Authors · 2023-11-20
> > **Response to Reviewer YRyF (2/2)**
> >
> > **Q4-1: Unclear phrasing regarding "three implementations" of UniVis.**
> > > At p. 3, you talk about "three implementations" of UniVis. But I would hesitate to call these different implementations. I think the contribution of your paper is that these are all the same implementation, but simply trained in three different regimes (single-task, single-category, and multi-category). I prefer the phrase you use later: "three types of model training."
> >
> > **A4-1:** We have changed "three implementations of UniVis" to "three types of UniVis trained in different regimes" (at the top of page 3).
> >
> > **Q4-2: A typo.**
> > > In the first sentence of the abstract: the word "tam" seems to be a typo.
> >
> > **A4-2:** Thanks for pointing this out. It should be "tame". We are not allowed to make changes to the title and abstract during rebuttal per ICLR policy. We will fix this in the final version if our paper gets accepted.
> >
> > **Q4-3: Unclear phrasing regarding "all-rounder".**
> > > In the introduction, "all-rounder" is unclear.
> >
> > **A4-3:** We have changed "all-rounder" to "generalist" (see first paragraph of Section 1) as the latter would be more well-understood.
> >
> > **Q4-4: Awkward phrasing in the introduction.**
> > > In the introduction, the sentence "The Challenges are in three aspects, elaborated in the following" is an awkward sentence. I suggest simply: "There are three main challenges."
> >
> > **A4-4:** Following your suggestion, we have simplified that sentence in the revised paper (see second paragraph of Section 1).
> >
> > **Q5: Unclear definition of "best" and "second best".**
> > > Why do you ignore the apparently better-performing comparator models in several of the tables when reporting the "best" and "second best"? For instance, in Table 1, why does OneFormer not get bolded as the "best"? If you are ignoring specialized models in your ranking of best and second best, you should include this caveat in your description of "best."
> >
> > **A5:** We have updated the table captions (see Table 1) to include a more precise definition of "best" and "second best" as follows: "We ignore specialized models when ranking best and second best and this applies to all tables."
> >
> > **Q6: UniVis-sc on other categories of vision tasks.**
> > > I see that you trained a single-category UniVis on the four conditional image generation tasks. Did you attempt training your multi-task/single-category UniVis on either of the two other categories? If not, why did you choose conditional image generation as the category to try as the single-category UniVis?
> >
> > **A6:** We also trained a single-category UniVis on three low-level image processing tasks. We have updated Table 2 to add the corresponding results. This demonstrates the validity of UniVis-sc in handling different low-level image processing tasks using a shared weight.
> >
> > **Q7: UniVis-mc on other combinations of tasks.**
> > > I see that for multi-category UniVis, you selected depth estimation, denoising, and mask-to-image. Did you attempt training other combinations of tasks across the categories (e.g. semantic segmentation + deraining + pose-to-image)? If not, why did you choose the three you chose?
> >
> > **A7:** We chose depth estimation, denoising, and mask-to-image, in the sense that they involve disparate visual signals and data domains. We hope the results provided in Table 4 can verify the effectiveness of UniVis-mc under such challenging scenarios. Due to limited time during rebuttal, we are unable to train a new model on another combination of tasks across the categories, but we promise to add the experimental results to the final version if the paper gets accepted.
> >
> > **Q8: Difficult cases of joint training.**
> > > Do you expect some combinations of tasks to be particularly difficult for UniVis to be trained for at the same time?
> >
> > **A8:** We did not observe such situations in our experiments. Maybe keypoint detection and depth-to-image generation are difficult for UniVis to be jointly trained as the former task mainly focuses on detecting sparse keypoints for humans but the latter aims at translating a dense depth map into a realistic image.

---

> > > ### Author Response · Authors · 2023-11-22
> > > **Official Comment by Authors**
> > >
> > > Dear Reviewer YRyF,
> > >
> > > We sincerely appreciate your valuable feedback on our submission. As the author-reviewer discussion nears its end, we are eager to know if our responses have addressed your concerns. We are also more than willing to engage in further discussions if needed.
> > >
> > > Thank you again for your time and effort!

---

> > > > ### Comment · Reviewer_YRyF · 2023-11-23
> > > > **I have more confidence in the evaluations**
> > > >
> > > > The newly reported standard deviations help provide necessary context for anyone trying to understand the significance (in a collquial sense even if not in a statistical sense) of the reported performance numbers. Thank you. I understand you have promised to add some measure of deviation to all tables for the final version. When you do, please explain somewhere (perhaps in the first table, or in the methodology section) how these +- figures are obtained (is it std dev, variance, a confidence interval, how many reps, etc.) *This raises my soundness score to 4*.
> > > >
> > > > Your edits also greatly clarified the points I thought could be better explained. *This raises my presentation score to 4*.
> > > >
> > > > I disagree that a novel approach to a generalizable multi-task/category model must be shown to outperform other single-task methods out of the gate. Reviewer bP4y asks: "why not select the most effective method for each specific task and combine them to be a 'universal solver'"? I understand the desire to continually seek strictly higher performance measures, and of course, that might be the appropriate pragmatic approach if one were deploying a live solution in a product today, but novel methods don't need to beat state-of-the-art to make useful contributions to the field. Nor do I think the method would have to exhibit "emergent" properties.

---

> > > > > ### Author Response · Authors · 2023-11-23
> > > > > **Official Comment by Authors**
> > > > >
> > > > > We sincerely appreciate your acknowledgment of our rebuttal and kind words to root for our paper.
> > > > >
> > > > > We have revised our paper again to add an explanation of these +- figures in the caption of Table 1 as follows: "The results of UniVis are reported as the average scores and standard deviations across three trials." We will update all tables to include this measure of deviations in the final version. Thank you.

---

### Official Review · Reviewer_bP4y · 2023-11-04

**Soundness:** 2 fair
**Presentation:** 3 good
**Contribution:** 1 poor
**Rating:** 3
**Confidence:** 4

**Summary:**

This paper proposes UniVis, a framework that can deal with several visual tasks, including visual understanding (e.g., semantic segmentation), low-level image processing (e.g., denoising), and conditional image generation. The idea is to perform instruction tuning on a large-scale pre-trained text-to-image diffusion model. During training, the model is trained to fill in the missing query output based on the provided instructions, which consists of task-specific input-output pairs.

**Strengths:**

This paper proposes a unified framework for multiple common visual tasks across different categories, including image understanding, image processing, and image generation. The idea of using instruction tuning on Stable Diffusion is novel and interesting, and the experimental results are reasonable. The presentation of the paper is also clear.

**Weaknesses:**

1. The proposed method is limited to dense image tasks, where the output is at a high-dimensional image level. It does not demonstrate feasibility on a wide range of computer vision tasks where the output is low-dimensional, such as image classification, image captioning, VQA, etc.
2. Even for dense image tasks, it misses multiple tasks. For example, for the task of conditional image generation, the paper misses two important tasks: class-conditional generation and text-conditional generation. Thus, I feel the claim that Univis is a "Universal Framework for Computer Vision Tasks" is exaggerated, making the contribution of the paper limited.
3. Overall, the experiment results shown in the paper are weak. In two of the three categories, it is far behind the Painter baseline. It might be because of computation limitations, but it also might be because the proposed method does not scale with more computation resources. Therefore, it would be good to show at least one experiment that is trained with the same computation budget as the Painter baseline, otherwise, the results are not convincing enough to demonstrate any improvement over previous methods.
4. The method does not show scaling ability with more tasks and data. As shown in Table 4, a model trained with multiple tasks is worse than a model trained on a single task. This raises concerns that this method may not scale up well with multi-task training, which is not desirable in a universal framework.

**Questions:**

Please see weaknesses

---

> ### Author Response · Authors · 2023-11-20
> **Response to Reviewer bP4y**
>
> Thank you for the insightful review. Below, we answer all the questions.
>
> **Q1: Limited to dense image tasks.**
> > The proposed method is limited to dense image tasks, where the output is at a high-dimensional image level. It does not demonstrate feasibility on a wide range of computer vision tasks where the output is low-dimensional, such as image classification, image captioning, VQA, etc.
>
> **A1:** (1) We have evaluated UniVis on keypoint detection, where the output is low-dimensional (please refer to Figures 3, 8, and 17). (2) Other computer vision tasks with sparse outputs could be tackled by our framework as long as one can transfer the output to RGB images. Taking image classification as an example, a class label can be turned into an RGB image filled with a unique color determined by its semantic class. Due to limited time during rebuttal, we cannot provide the results on image classification at this time, but we promise to include them in the final version if the paper gets accepted. (3) We would like to clarify that this paper focuses on standard computer vision tasks instead of vision-language ones including image captioning and VQA, which are also not explored by our competing methods (Painter and PromptDiffusion).
>
> **Q2: More dense image tasks.**
> > Even for dense image tasks, it misses multiple tasks. For example, for the task of conditional image generation, the paper misses two important tasks: class-conditional generation and text-conditional generation. Thus, I feel the claim that Univis is a "Universal Framework for Computer Vision Tasks" is exaggerated, making the contribution of the paper limited.
>
> **A2:** (1) Text-conditional generation can be fulfilled by directly applying UniVis-sc trained on four conditional image generation tasks where the query is set to a black image. Please see Figure 16 in the revised paper for visual results and more details in the updated Appendix (Sec. C). (2) Class-conditional generation could also be achieved by framing it as an inverse task of image classification (just like depth-to-image generation and depth estimation). We promise to provide the results on class-conditional generation in the final version if the paper is accepted. (3) We believe that a wide range of vision tasks from three **distinct** categories evaluated in our paper would adequately support our major claims.
>
> **Q3: Weak performance on visual understanding and low-level image processing tasks.**
> > Overall, the experiment results shown in the paper are weak. In two of the three categories, it is far behind the Painter baseline. It might be because of computation limitations, but it also might be because the proposed method does not scale with more computation resources. Therefore, it would be good to show at least one experiment that is trained with the same computation budget as the Painter baseline, otherwise, the results are not convincing enough to demonstrate any improvement over previous methods.
>
> **A3:** Please see our general response #1. The results provided in the second table in general response #1 verify that UniVis scales well with more computation resources.
>
> **Q4: Scaling ability with more tasks and data.**
> > The method does not show scaling ability with more tasks and data. As shown in Table 4, a model trained with multiple tasks is worse than a model trained on a single task. This raises concerns that this method may not scale up well with multi-task training, which is not desirable in a universal framework.
>
> **A4:** We would like to clarify that our primary focus is to investigate how to induce a profound understanding of vision tasks (which involve very disparate visual signals and data domains) through a shared scheme of **generative modeling** instead of seeking gains with multi-task training. Please see our general response #2 for more details.

---

> > ### Author Response · Authors · 2023-11-22
> > **Official Comment by Authors**
> >
> > Dear Reviewer bP4y,
> >
> > We sincerely appreciate your valuable feedback on our submission. As the author-reviewer discussion nears its end, we are eager to know if our responses have addressed your concerns. We are also more than willing to engage in further discussions if needed.
> >
> > Thank you again for your time and effort!

---

> ### Comment · Reviewer_bP4y · 2023-11-22
>
> Thank you for the authors' rebuttal and the clarification regarding the major aim of the paper. After reviewing the rebuttal, I have reassessed my original score, leading to its reduction. This decision stems from the authors' statement that the main goal of the paper is to "reveal the potential of generative modeling in building a universal solver for vision tasks", while not "seeking gains with multi-task training". This perspective notably diminishes the significance of a 'universal solver'. If UniVis does not have the ability to improve the performance with more tasks, a basic 'universal solver' might simply be a combination of Painter + PromptDiffusion, which shows better performance across all three evaluated tasks than UniVis.
>
> In short, if a combined approach (UniVis-mc) does not elevate performance in any of the individual tasks (UniVis-st), then the benefit of this combination is quite constrained. It prompts the question: why not select the most effective method for each specific task and combine them to be a "universal solver"? One minor benefit of a universal solver with poor performance in all tasks could be a reduction in the total number of parameters required, but this aspect is not addressed in the paper.

---

> > ### Author Response · Authors · 2023-11-23
> > **Clarification by Authors**
> >
> > Thanks for your feedback. We gracefully disagree with some of your comments and would like to provide some clarifications as follows：
> >
> > 1. The "universal" essence of our UniVis lies in the **shared** framework for a diverse set of vision tasks, and its significance stems from revealing the potential of the unifying capability of a single generative model, which was not investigated before.
> > 2. If "a basic 'universal solver' for UniVis might simply be a combination of Painter + PromptDiffusion", one could say that a basic "universal solver" for Painter would be a combination of expert models for each task, which shows better performance than Painter, then what is the point of Painter (and PromptDiffusion)? We believe **minimizing the inductive bias inherent in designing models for individual tasks is important and meaningful**, which is what Painter, PromptDiffusion, and the proposed UniVis all seek.
> > 3. Painter shows that joint training of prediction and low-level image processing tasks brings performance gains on most of the tasks, and we also verify this via joint training of generation tasks (see Table 3) and joint training of low-level image processing tasks (see Table 2 in the revised paper). We do not have as many computing resources as Painter to train a single model involving all tasks, but we tried our best to showcase that UniVis-mc can work WELL on three representative tasks from different categories. This reveals the ability of UniVis in **simultaneously handling discrimination and generation**, which is beyond the reach of prior methods.
> >
> > We are open to any further discussion and would appreciate a reassessment of our work. Thank you.

---

### Author Response · Authors · 2023-11-20
**General Response to All Reviewers (1/2)**

We sincerely thank all reviewers for recognizing that our paper is well-written and clearly presented (Reviewer bP4y, ZgiD), has novel, interesting/intriguing, and original ideas (Reviewer bP4y, YRyF, ZgiD), has thorough evaluation and reasonable/competitive results (Reviewer bP4y, ZgiD, zYeV), and showcases the unifying capability of UniVis (Reviewer YRyF, ZgiD, zYeV). We also appreciate their constructive comments and suggestions, which genuinely help improve our paper. **Revisions are colorized (in blue) in the new version of our paper.**

We identify two primary concerns shared by reviewers and would like to address them as follows:

### **#1. Weak performance on visual understanding and low-level image processing tasks**

First, Painter uses much larger computing resources (32$\times$) than us and we can access only academia-level resources (4 RTX 3090 GPUs). When using the same computing resources, the proposed UniVis attains a competitive performance close to Painter on visual understanding and low-level image processing tasks. It is also worth noting that there is an *upper bound* for UniVis on low-level image processing because the autoencoder of pre-trained SD brings information loss (please refer to the caption of Table 2). Most importantly, as shown in Tables 1 and 3, both Painter and PromptDiffusion experience a clear collapse or near breakdown on one of the three categories of vision tasks. For instance, on the conditional image generation tasks, Painter completely collapses while UniVis exhibits SOTA performance (see Table 3 and Figures 12, 13, 14, and 15). In other words, UniVis could handle at least one more category of vision tasks compared to its competitors (which was also discussed in Sec. 5 of the main paper). We conclude this in the following table.

|      Method     | Visual Understanding | Low-level Image Processing | Conditional Image Generation |
|:---------------|:--------------------:|:--------------------------:|:----------------------------:|
|     Painter     |       &#10004;       |          &#10004;          |           &#10008;           |
| PromptDiffusion |       &#10008;       |          &#10004;          |           &#10004;           |
|      UniVis     |       &#10004;       |          &#10004;          |           &#10004;           |

For a more intuitive comparison, we draw a radar chart (**please see Figure 5 in the revised paper**) to showcase the overall performance of different methods on three types of tasks. UniVis achieves the most balanced and comprehensive performance.

Second, we try our best to conduct an experiment where UniVis is trained using different amounts of computing resources. We show the results below. It indicates **the scaling ability of our UniVis with larger computing power.** We hope this is convincing that UniVis is very likely to get comparable performance to Painter on visual understanding/low-level image processing tasks by scaling up the computation resources.

|                         |   |                 | Depth estimation |        |   |       | Denoising |       |
|:-----------------------|---|:---------------:|:----------------:|:------:|---|:-----:|:---------:|:-----:|
| **Computing resources** |   | RMSE$\downarrow$ |        REL$\downarrow$       | $\delta_{1}$$\uparrow$ |   |  PSNR$\uparrow$ |    SSIM$\uparrow$   | LPIPS$\downarrow$  |
|       one 3090 GPU      |   |      0.461      |       0.156      |  0.812 |   | 34.02 |   0.898   | 0.121 |
|      four 3090 GPUs     |   |      0.420      |       0.135      |  0.857 |   | 34.55 |   0.907   | 0.095 |
|     eight A100 GPUs     |   |      0.391      |       0.118      |  0.892 |   | 34.92 |   0.913   | 0.092 |

At last, we would like to emphasize that the major aim of our paper is to reveal the potential of **generative modeling** in building a universal solver for vision tasks and spur further exploration into this, with achieving state-of-the-art performance being of our lower priority.

---

> ### Author Response · Authors · 2023-11-20
> **General Response to All Reviewers (2/2)**
>
> ### **#2. Joint training does not yield significant gains**
> As shown in Table 4, UniVis-mc exhibits a competitive performance very close to UniVis-st. UniVis-sc produces slightly better results than UniVis-st on most of the tasks (see Tables 2 and 3). These results indicate that the proposed framework has a large potential to unify various tasks including discrimination and generation which are typically studied exclusively, which is indeed the aim of our work and beyond the reach of competing methods (see first table in general response #1). Seeking gains with multi-task training is not our focus (we do not explore any multi-task training skills as well, especially under limited computing resources).
>
> As pointed out by Reviewer YRyF, the ability to produce a single trained model that can generalize across diverse categories of computer vision tasks by simply altering the "instruction" would be very **useful**. We also showcase the generalization capability of UniVis in Figure 4 and Figure 17 (**newly added to the revised paper**). We hope these merits of UniVis would encourage further investigation on how to induce a profound understanding of vision tasks (which involve very disparate visual signals and data domains) through a shared scheme of **generative modeling**.
>
> We address all other concerns in individual replies to the reviewers.

---

### Author Response · Authors · 2023-11-23
**Last Response to AC and Reviewers**

We want to give a brief message at the end of the author-reviewer discussion phase as follows:

We believe the proposed UniVis makes significant contributions to the field of multi-task/category vision models, as it reveals the capability of a single generative model in unifying diverse tasks, including discrimination and generation. As pointed out by Reviewer YRyF, it DOES NOT require SOTA performance on every task to claim so.

Thank you for your time.

---

### Meta-Review · Area_Chair_38T9 · 2023-12-05

**Metareview:**

This paper presents a universal learning framework that combines vision and language and can be applied to different vision tasks.

After the rebuttal and AC-reviewer discussion stage, the final scores of this paper are 3/5/5/8. All reviewers responded in the discussion stage and none of them changed the score. The positive reviewer (rating 8) said that he/she would not defend this paper, although he/she believed that this is a good paper.

The AC also read the paper and here are the comments.

This paper has a big picture, i.e., to unify as many vision tasks as possible into the same framework. It followed the in-context learning framework which itself was not completely new (the authors are aware of this and cited some prior works such as Painter). The key difference lies in the use of a text prompt to assist the task. But, I think this also brings a negative impact, especially in the true ability of unification.

For example, to enable image generation, the text prompt must be specifically designed for the specific input image (e.g., as shown in Figure 1, "a half eaten sandwich on top of a white plate"). This mechanism, more or less, harms the generalization ability. It also makes me doubt whether the performance gain over other methods (e.g. Painter) is brought by the additional text prompt.

In summary, this paper seems to deliver an over-optimistic message to the community, but what it has truly done does not support its statement. Therefore, the AC chooses to agree with the majority of reviewers to recommend rejection.

**Justification For Why Not Higher Score:**

The reviewers arrived at a consensus of rejection.

**Justification For Why Not Lower Score:**

N/A

---

### Decision · Program_Chairs · 2024-01-16

Reject